



# Cloud liquid water path detectability and retrieval accuracy from airborne passive microwave observations over Arctic sea ice

Nils Risse[1], Mario Mech[1], Catherine Prigent[2], Joshua J. Müller[3], and Susanne Crewell[1]

[1]Institute for Geophysics and Meteorology, University of Cologne, Cologne, Germany
[2]Laboratoire d'Instrumentation et de Recherche en Astrophysique, Observatoire de Paris, CNRS, Paris, France
[3]Leipzig Institute for Meteorology, University of Leipzig, Leipzig, Germany

**Correspondence:** Nils Risse (n.risse@uni-koeln.de)

**Abstract.** Clouds are critical in the Arctic's water balance and energy budget. Especially, the cloud liquid water path (CLWP) modifies the cloud radiative properties and affects the surface energy balance. Spaceborne microwave radiometers provide a high sensitivity to CLWP at pan-Arctic scales, but extracting this information over sea ice requires separation of surface and cloud emission. Here, we assess CLWP detectability and retrieval accuracy over sea ice from a physical optimal estimation retrieval applied to airborne passive microwave observations during the HALO–$(\mathcal{AC})^3$ campaign. Reference data on surface temperature, young ice fraction, hydrometeor occurrence, and cloud liquid layers are available from collocated airborne instruments. The retrieval estimates CLWP and five surface parameters by inverting a forward operator consisting of the Snow Microwave Radiative Transfer (SMRT) and Passive and Active Microwave radiative TRAnsfer (PAMTRA) models. We find a consistent representation of sea ice and snow emission from 22–183 GHz under clear-sky conditions in both observation and state space. The CLWP detectability, defined as the 95th percentile of retrieved CLWP under clear-sky conditions, is about $50\,\mathrm{g\,m^{-2}}$ in the Central Arctic and increases towards the marginal ice zone up to $350\,\mathrm{g\,m^{-2}}$. The CLWP retrieval accuracy increases with increasing CLWP, with a relative root mean squared error below $50\,\%$ for CLWP above $100\,\mathrm{g\,m^{-2}}$. Retrieval uncertainties occur due to ambiguities between cloud liquid water emission and scattering in the snowpack and emission by newly formed sea ice. We further analyze the impact of surface melt and a rain-on-snow event associated with the warm air intrusion on the surface parameters. Finally, we show CLWP distributions along the flight track for all HAMP observations in comparison to ERA5 for different cloud regimes. The retrieval algorithm enhances the understanding of Arctic clouds and allows for an improved use of passive microwave satellite data in polar regions.

## 1   Introduction

The Arctic is warming at a faster rate than the global average in recent decades (Rantanen et al., 2022). Clouds play a critical role as a feedback mechanism in the amplified warming in the Arctic (Tan et al., 2021). Cloud liquid water modifies the cloud radiative effect (Shupe and Intrieri, 2004; Ebell et al., 2020) with implications for the surface energy budget (Sledd et al., 2025). Additionally, cloud liquid water plays a critical role in precipitation formation processes, such as efficient ice crystal growth through riming (Maherndl et al., 2024). Passive microwave radiometers allow for a quantification of the cloud liquid water path (CLWP), defined as the columnar integral of cloud liquid water content, by observing the temperature-dependent



microwave emission of liquid droplets (Kneifel et al., 2014; Turner et al., 2016). This emission of liquid droplets increases
with frequency, and retrievals typically use observations at window channels in the range from 19 to 90 GHz (Greenwald
et al., 1993; Crewell and Löhnert, 2003). Over sea ice, high-quality CLWP estimates are provided from ship-based microwave
radiometers (Westwater et al., 2001; Walbröl et al., 2022). Operational satellite CLWP products are currently not available
over sea ice due to the variable emission and polarization of the sea ice and snow. For example, the Multisensor Advanced

Climatology of Liquid Water Path (MAC-LWP;  Elsaesser et al., 2017; O'Dell et al., 2008) is limited to ice-free ocean, and
the Microwave integrated Retrieval System (MiRS;  Boukabara et al., 2011) provides estimates over ice-free ocean and land
only. First estimates of the CLWP retrieval accuracy from passive microwaves are presented by Haggerty et al. (2002) using
airborne microwave observations and collocated in situ data. Their results show a high accuracy for CLWP above $100\,\mathrm{g\,m^{-2}}$
and poor accuracy for CLWP below $50\,\mathrm{g\,m^{-2}}$. Generally, the CLWP uncertainty increases with increasing surface emissivity

due to the decreasing contrast between the liquid cloud emission and the surface (Prigent et al., 2003). Hence, airborne or
satellite retrievals of atmospheric properties require an accurate representation of the surface emissivity and its polarization.

Several methods were developed to describe the emissivity of sea ice and snow-covered surfaces. Emissivity atlases derived
from long-term satellite observations under clear-sky with collocated surface temperature data provide robust first-guess emis-
sivities and their variability (Prigent et al., 1997; Wang et al., 2017). However, as the emissivity variability over sea ice can

be very high, the long-term mean might deviate largely from the actual emissivity (Perro et al., 2020). To better capture this
emissivity variability, dynamic emissivity modeling approaches are developed in a data assimilation context (e.g., Di Tomaso
et al., 2013). This approach computes the emissivity at window channels and extrapolates to neighboring sounding channels
in the same field-of-view, but is limited to clear-sky conditions. A novel machine learning approach by Geer (2024) addresses
the need for better sea ice emissivity modeling in a numerical weather prediction context over sea ice (Lawrence et al., 2019).

The approach exploits long-term observations to learn a compact representation of relevant sea ice and snow microphysical
properties and their empirical transformation to an emissivity. While machine learning provides a computationally efficient
approach, interpreting the underlying geophysical parameters is challenging. Physical snow and sea ice radiative transfer mod-
eling approaches directly compute the emission from plane parallel sea ice and snow layers and their properties, such as density,
grain size, salinity, temperature, thickness, and microstructure (Tonboe et al., 2006). The Arctic-wide retrieval by Rückert et al.

(2023b) validated with observations from the Multidisciplinary drifting Observatory for the Study of Arctic Climate (MO-
SAiC) expedition provides simultaneous atmospheric and sea ice properties from an optimal estimation retrieval framework.
While their retrieval used a fixed assumption on the snow and ice layering, Kang et al. (2023) explored the coupling of a sea
ice and snow radiative transfer model with a thermodynamic sea ice and snow evolution model to better capture snow meta-
morphism and temporal variations in snow layering. This approach could be useful in a coupled land-atmosphere assimilation

of surface-sensitive microwave channels (Hirahara et al., 2020). Yet, we lack a detailed assessment of the CLWP detectability
and retrieval accuracy from passive microwave observations over sea ice.

To study the CLWP signal over sea ice, we develop an optimal estimation sea ice–atmosphere retrieval specifically for
airborne passive microwave observations from 22–183 GHz at nadir. The underlying forward operator simulates the brightness
temperature ($T_b$) at flight altitude from a loose coupling of the Snow Microwave Radiative Transfer (SMRT; Picard et al., 2018)





model with the Passive and Active Microwave radiative TRAnsfer (PAMTRA; Mech et al., 2020) tool via the spectral surface emissivity and effective temperature. This physical modeling approach allows a simultaneous retrieval of snow layer properties (correlation length and thickness), snow and sea ice temperature, and CLWP, under non-heavy cloud ice and snow conditions, since frozen water path is not retrieved. We apply the retrieval to the airborne Microwave Package (HAMP; Mech et al., 2014) radiometer onboard the *High Altitude and Long Range Research Aircraft* (*HALO*) during the HALO–$(\mathcal{AC})^3$ field campaign carried out in spring 2022 in the Fram Strait and Central Arctic. *HALO's* cloud observatory suite (Stevens et al., 2019) with coincident cloud radar, lidar, and infrared observations provides a unique opportunity for passive microwave retrieval evaluation (Jacob et al., 2019). We aim to (1) assess the representation of sea ice and snow microwave emission by the forward model, (2) estimate the CLWP detectability and retrieval accuracy, and (3) analyze the spatial variability of CLWP over sea ice during HALO–$(\mathcal{AC})^3$.

The paper is structured as follows. Section 2 provides an overview of the airborne field data and auxiliary satellite and reanalysis data. Section 3 describes the sea ice–atmosphere retrieval and the forward operator. Section 4 details the clear-sky evaluation (first objective), CLWP detectability, and CLWP retrieval accuracy (second objective). Section 5 addresses the third objective by presenting the retrieval application to two case studies, a rain-on-snow event, and comparing the CLWP from HAMP with ERA5. The study is summarized and concluded in Sect. 6.

## 2 Data

### 2.1 HALO–$(\mathcal{AC})^3$ field campaign

The multi-platform field campaign HALO–$(\mathcal{AC})^3$ included 17 flights with the research aircraft *HALO* between 11 March and 12 April 2022 over sea ice in the Fram Strait and Central Arctic (Wendisch et al., 2024; Ehrlich et al., 2024). Thus, *HALO* captured diverse sea ice conditions from young ice near the sea ice edge to perennial sea ice north of Greenland. Here, we include all observations over at least 90 % sea ice concentration (Spreen et al., 2008) with a distance of more than 15 km to coasts (Fig. 1). Due to the coarser spatial resolution of the sea ice product compared to the airborne observations, few open water pixels remain in the airborne data.

The meteorological conditions during HALO–$(\mathcal{AC})^3$ were dominated by warm air intrusions from 11–20 March 2022 and colder northerly winds from 21 March 2022 until the end of the campaign (Walbröl et al., 2024). The warm air intrusions caused rainfall on sea ice up to about 83° N (see Fig. 10 in Walbröl et al., 2024), which *HALO* captured on three consecutive days (11–13 March 2022).

The cloud observatory configuration of *HALO* includes a microwave radiometer, cloud radar, lidar, thermal infrared radiometer, thermal infrared spectral imager, and solar spectral imager. In addition, 85 dropsonde launches over sea ice provide vertical profiles of air temperature, humidity, and wind between flight altitude and the surface (George et al., 2024). This dropsonde data was partly assimilated into the European Centre for Medium-Range Weather Forecasts (ECMWF) Integrated Forecasting System (IFS). Details on the *HALO* instrumentation and dropsonde assimilation can be found in Ehrlich et al. (2024). An





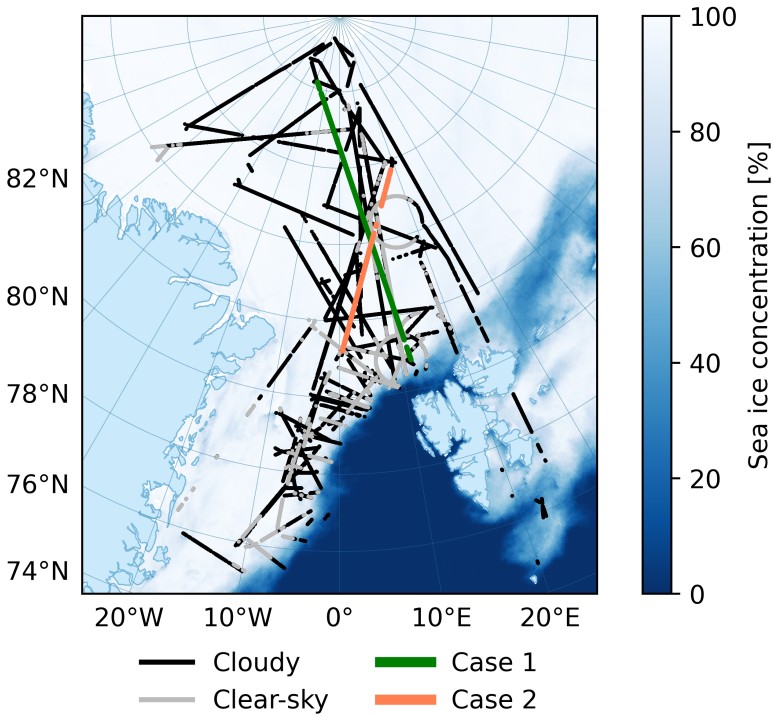

**Figure 1.** Map of the *HALO* flight track and mean sea ice concentration (Spreen et al., 2008). Only positions where the retrieval was applied are shown.

example of the microwave radiometer, radar, and lidar observations is provided in Fig. 2 for a 550 km southbound transect. The following sections describe the instruments and products of the cloud observatory configuration and ancillary products.

## 2.2 Microwave radiometer

95 The HAMP radiometer measures at 25 channels in the frequency range from 22.24 to 183.31±7.5 GHz (Mech et al., 2014). Six channels each are located along the 22.24 GHz water vapor absorption line and around the 183.31 GHz water vapor absorption line, seven channels are located along the 50–60 GHz oxygen absorption complex, four channels are located around the 118.75 GHz oxygen absorption line, and two channels are located within atmospheric windows at 31.4 and 90 GHz. HAMP points nadir and samples with a temporal resolution of 1 s. The footprint sizes range from about 0.7 to 1.4 km at typical *HALO*

100 flight altitude and speed (Table 1). The data was corrected for biases using dropsondes over open ocean (Dorff et al., 2024). Here, we use an updated version of the bias correction. Measurement gaps that were filled by temporal interpolation in the published data are discarded, and we removed any observations where aircraft roll or pitch exceed ±6°. Moreover, we exclude about 19 % of the observations due to potentially high scattering by frozen hydrometeors or surface melt. In total, about 85,000



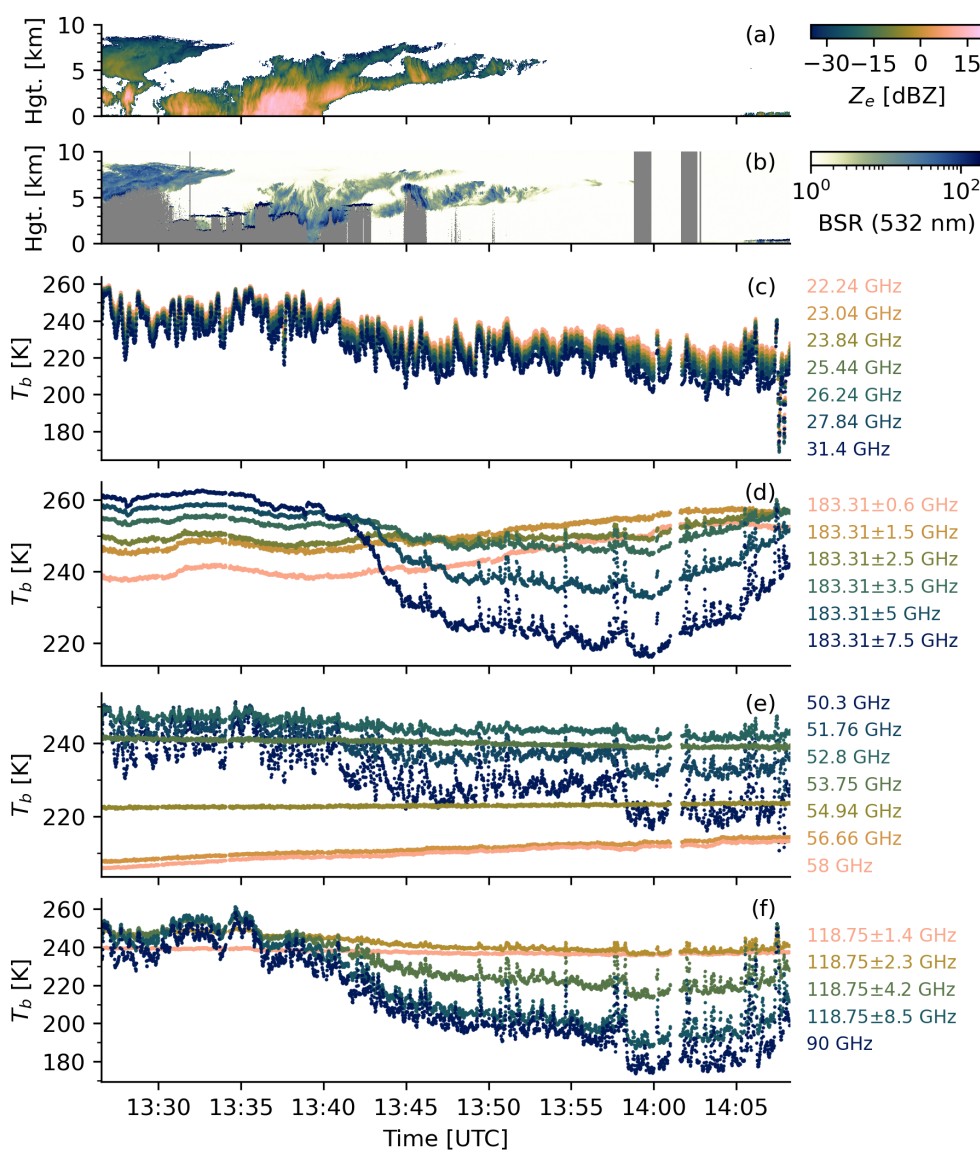

**Figure 2.** Radar, lidar, and microwave radiometer observations during a 550 km southbound transect over sea ice on 14 March 2022 (case 2 in Fig. 1). (a) Radar reflectivity, (b) lidar backscatter ratio, (c) $T_b$ from 22 to 31 GHz, (d) $T_b$ around 183.31 GHz, (e) $T_b$ from 50 to 58 GHz, and (f) $T_b$ at 90 and around 118.75 GHz. Missing/flagged data is shown in gray.





**Table 1.** Beam width, noise equivalent differential temperature (NeDT), and footprint size of HAMP channels. The footprint size is calculated for a flight velocity of $300\,\mathrm{m\,s^{-1}}$ and 12 km flight altitude.

| Channels | Frequency range [GHz] | Beam width [°] | NeDT [K] | Footprint size [km$^2$] |
|---|---|---|---|---|
| 1–7 | 22.24–31.4 | 5 | 0.1 | 1.1×1.4 |
| 8-14 | 50.3–58 | 3.5 | 0.2 | 0.7×1 |
| 15 | 90 | 3.3 | 0.25 | 0.7×1 |
| 16–19 | 110.25–127.25 | 3.3 | 0.6 | 0.7×1 |
| 20–25 | 175.81–190.81 | 2.7 | 0.6 | 0.6×0.9 |

HAMP samples are available over sea ice along a flight distance of 20,000 km between -55–27° E and 74.8–89.4° N, out of which about 14 % (12,200) were clear-sky as identified from the radar–lidar cloud mask with available thermal infrared data.

Most weighting functions of HAMP peak at the surface under cold and dry Arctic conditions. Thus, the $T_b$ varies due to changes in surface emission along the flight track with a high correlation between neighboring surface-sensitive channels (Fig. 2c–f). Therefore, we only use six HAMP channels for the retrieval: 22.24, 31.4, 50.3, 90, 118.75±8.5, and 183.31±7.5 GHz. This includes channels typically used for ground-based and satellite CLWP retrievals, with the highest sensitivity at 90 and 118.75±8.5 GHz. Moreover, these channels fully exploit HAMP's spectral range for surface characterization.

## 2.3 Radar–lidar cloud mask

The cloud radar and lidar onboard *HALO* provide reference data on the occurrence of hydrometeors in the field of view of HAMP. Especially, the lidar is highly sensitive to liquid cloud layers. Both instruments and derived products are described below.

The HAMP cloud radar operates in the Ka-band at 35.5 GHz with a temporal resolution of 1 s and vertical resolution of 30 m (Ewald et al., 2019). The sensitivity of the HAMP radar is about -30 dBZ (Konow et al., 2019). Here, we use the radar reflectivity product aligned temporally with the passive microwave radiometer observations and filtered for ground clutter in the lowest about 100 m (Dorff et al., 2024). Hence, shallow fog layers cannot be detected by the radar. Compared to the microwave radiometer, the radar's footprint size is rather narrow, with about 130 m (Mech et al., 2014).

Backscatter lidar and water vapor differential absorption lidar profiles were measured by the airborne demonstrator for the WAter vapor Lidar Experiment in Space (WALES; Wirth et al., 2009). Here, we use the backscatter ratio (BSR) and depolarization ratio at 532 nm, which are available with a vertical resolution of 15 m and a temporal resolution of 1 s (Wirth and Groß, 2024). We exclude all data with a non-zero quality flag and below 50 m above the surface.

The radar observation is defined as cloudy if the radar reflectivity of any bin exceeds -40 dBZ. Similarly, we apply a backscatter ratio threshold of 4 to the lidar column. We define a scene as cloudy if either the radar or the lidar observations fulfill their cloud mask criterion. Both thresholds reduce the impact of thin ice clouds on the thermal infrared radiometer measurements.



In addition to the hydrometeor detection, we need to identify scenes with potential impact of scattering by frozen hydrometeors, which is relevant at HAMP frequencies above 90 GHz (Bennartz and Bauer, 2003). Here, we use a maximum radar reflectivity threshold of 5 dBZ at any height level in the radar column, which corresponds to a snowfall rate of about 0.05 to 130   0.5 mm h$^{-1}$ depending on the ice particle habit and size distribution (Kneifel et al., 2011).

Further, we build a detection method for liquid cloud layers based on the lidar backscatter ratio and depolarization ratio. Cloud regions dominated by liquid water exhibit a high backscatter and near-zero depolarization ratio (Shupe, 2007; De Boer et al., 2009; Luke et al., 2010). Several threshold-based methods are developed for liquid classification from both parameters (Kalesse-Los et al., 2022), and here we subjectively define a similar thresholding method from the examination of WALES 135   statistics of both parameters for HALO–$(\mathcal{AC})^3$. We define a region as liquid-dominated if the depolarization ratio is below 0.1 and the backscatter ratio is above 50. Typically, only the uppermost liquid layer can be detected from airborne lidars, and we define the uppermost bin of liquid-dominated regions as liquid layer top height ($h_l$). To account for attenuation of the lidar beam by large amounts of frozen hydrometeors, we classify columns that did not satisfy the liquid water criterion as potentially liquid clouds if the radar hydrometeor fraction in the lowest 5 km exceeds 50 %.

## 2.4   Radiation data

The thermal infrared radiometer KT-19 provides $T_b$ in the atmospheric window from 9.6–11.5 μm (Schäfer et al., 2022). The $T_b$ accuracy of KT-19 is about 0.5 K. The instrument points nadir with a beam width of 2.3°, which is comparable to the HAMP radiometer channels. The sampling frequency of 20 Hz is averaged to 1 Hz to match the HAMP radiometer sampling. We convert the clear-sky infrared $T_b$ to surface skin temperature under the assumption of an infrared emissivity of 0.995 (Høyer 145   et al., 2017; Thielke et al., 2022). Remaining atmospheric effects in the atmospheric window are considered to be negligible. This data is used as a data source for the development of the microwave-only retrieval.

The Video airbornE Longwave Observations within siX channels (VELOX) camera provides two-dimensional thermal infrared $T_b$ in the atmospheric window from 8.65–12 μm (Schäfer et al., 2022). The data are available at a temporal resolution of 1 s (Schäfer et al., 2023a). Here, we use the 10.74±0.39 μm channel (band 3) for qualitative information on spatial surface 150   temperature features. From each image, we extract the cross-track scan at nadir. This data is used as a visualization during case studies.

The VELOX-based clear-sky surface classification product groups each pixel into four surface types, i.e., open water, sea ice water mixture, thin sea ice, and snow-covered sea ice (Müller et al., 2025). The classification exploits spatial skin temperature variations of sea ice, snow, and open water with a spatial resolution of about 10×10 m$^2$. We derive the thin sea ice area fraction 155   within the microwave radiometer footprint from the high-resolution pixel-based classification. The accuracy of the thin sea ice classification, defined as the ratio of correct to total predictions, is approximately 70 % (Müller et al., 2025). We use this data for retrieval evaluation under clear-sky conditions.

The spectrometer of the Munich Aerosol Cloud Scanner (specMACS) measures two-dimensional fields of reflected spectral radiances from 0.4–2.5 μm (Ewald et al., 2016; Weber et al., 2024a). SpecMACS points nadir with a field of view of about 35° 160   and a temporal resolution of 30 Hz. Since the visible bands were not available during HALO–$(\mathcal{AC})^3$, we use the 1 μm near





infrared radiance for qualitative information on clouds and surface conditions. This data is used as a visualization during case studies.

## 2.5 Ancillary products

The ERA5 reanalysis provides hourly air temperature, pressure, and specific humidity on 137 model levels, and skin tem-
perature, 2 m air temperature, and total column liquid water on surface levels at a spatial resolution of 31 km resampled to
$0.25 \times 0.25°$ (Hersbach et al., 2020). Here, we use data from the ERA5 grid cells that are nearest in space and time to the *HALO*
flight track. The data is used as input for the retrieval and, in the case of the total column liquid water, for comparison with the
CLWP retrieved from HAMP. The 2 m air temperature is used to filter potential surface melt, which occurred during parts of
the warm air intrusion over sea ice. Here, we use a 2 m air temperature threshold of -1 °C.

Daily sea ice concentration maps from the University of Bremen with a $6.25 \times 6.25\,\text{km}^2$ resolution based on Advanced
Microwave Scanning Radiometer - 2 (AMSR2) 89 GHz observations (Spreen et al., 2008) are used to filter for observations
over sea ice. To include data close to the north pole not covered by the AMSR2 swath, we assume sea ice concentrations are
above 90 % in this area. Based on this data, we define the sea ice edge as the 50 % sea ice concentration contour and the Central
Arctic as a region with a distance of at least 200 km from the sea ice edge.

We use Level 1C $T_b$ data from channel 17 of the Special Sensor Microwave Imager/Sounder (SSMIS; Kunkee et al., 2008)
onboard the DMSP-F16 satellite to get qualitative information on the spatial $T_b$ variability around the *HALO* track (NASA
Goddard Space Flight Center and GPM Intercalibration Working Group, 2022). Channel 17 of SSMIS measures vertically
polarized $T_b$ at 91 GHz under an incidence angle of 53° with a footprint size of $9 \times 15\,\text{km}^2$.

## 3 Sea ice–atmosphere retrieval

### 3.1 Retrieval overview

For a coupled sea ice–atmosphere retrieval using a physical forward operator, we need to solve the radiative transfer of both
the cryosphere and the atmosphere. Unfortunately, no model exists that simultaneously solves the radiative transfer equations
of both spheres. Therefore, previous work on sea ice–atmosphere retrievals performed a loose coupling between the sea ice
and snow radiative transfer model and radiative transfer model for the atmosphere via $T_b$ or emissivity and emitting layer
temperature (e.g., Kang et al., 2023; Sandells et al., 2024). Here, we follow the same approach and loosely couple the radiative
transfer models SMRT (Picard et al., 2018) for the surface and PAMTRA (Mech et al., 2020) for the atmosphere. Both models
are called sequentially, and the surface radiative properties are provided to PAMTRA as frequency-dependent emissivity and
emitting layer temperature. This workflow is depicted in Fig. 3a and relevant input parameters are listed in Table 2.

   Using optimal estimation (Rodgers, 2000), we retrieve six state parameters from the HAMP observations, considering ob-
servation, forward model, and a priori uncertainties. The retrieved state parameters are CLWP and five surface parameters:
wind slab correlation length, depth hoar correlation length, wind slab thickness, snow–ice interface temperature, and air–snow



interface temperature. The selection of these state parameters is based mainly on two criteria. First, there should be high sensitivity to state parameter variations within the parameter uncertainty range at HAMP frequencies based on SMRT–PAMTRA simulations. Second, ambiguities in the radiometric sensitivity between state parameters should be minimized to ensure stable

retrieval convergence, i.e., correlations between columns of the Jacobian should be low. Note that CLWP is the only atmospheric parameter that gets retrieved. The selection of snow parameters is also motivated by Wivell et al. (2023), who found that varying wind slab correlation length, depth hoar correlation length, and wind slab thickness reproduces observed tundra snow emissivity spectra from 89 to 243 GHz.

     The main challenge for the retrieval is the lack of ground truth on the surface characteristics along the flight track. Therefore,

we define two retrievals, hereafter referred to as retrieval 1 (R1; Fig. 3b) and retrieval 2 (R2; Fig. 3c). R1 is only applied to clear-sky and retrieves four surface parameters using wind slab and depth hoar correlation length a priori from the literature and an a priori guess of the wind slab thickness. Additionally, the air–snow interface is taken from KT-19, and CLWP is fixed to $0\,\mathrm{g\,m^{-2}}$. The retrieved distribution for all clear-sky samples (mean and standard deviation) of wind slab correlation length, depth hoar correlation length, and wind slab thickness is then used as a priori for the microwave-only retrieval R2 for

both clear-sky and cloudy conditions to remove potential biases of the a priori values in R1. The clear-sky data used for the calibration covers most parts of the *HALO* study area and is therefore likely representative for cloudy scenes (Fig. 1).

### 3.2   Optimal estimation

During the optimal estimation retrieval, the state vector $\boldsymbol{x}$ is iteratively updated until an optimal solution is found. Here, we use a priori as a first guess. The updated state

$$\boldsymbol{x_{i+1}} = \boldsymbol{x_a} + \mathbf{S_i}\mathbf{K_i}^T\mathbf{S_e}^{-1}[(\boldsymbol{y} - F(\boldsymbol{x_i},\boldsymbol{b}) + \mathbf{K_i}(\boldsymbol{x_i} - \boldsymbol{x_a})] \tag{1}$$

is computed from the observation $\boldsymbol{y}$ (HAMP $T_b$), effective measurement uncertainty $\mathbf{S_e}$, a priori state $\boldsymbol{x_a}$, forward operator $F$, forward model parameters $\boldsymbol{b}$, Jacobian matrix $\mathbf{K_i}$ of the forward operator $F(\boldsymbol{x_i},\boldsymbol{b})$, and retrieval uncertainty $\mathbf{S_i}$. The effective measurement uncertainty combines observation and model uncertainty, i.e.,

$$\mathbf{S_e} = \mathbf{S_y} + \mathbf{K_b}\mathbf{S_b}\mathbf{K_b}^T, \tag{2}$$

with the observation uncertainty $\mathbf{S_y}$, the Jacobian matrix for model parameters $\mathbf{K_b}$ (computed during each iteration), and the model parameter uncertainty $\mathbf{S_b}$. We choose an uncorrelated observation uncertainty of 1.5 K from 22 to 118 GHz and 2 K at 183 GHz. Table 2 lists the mean and uncertainty of the model parameters (wind slab density, depth hoar density, depth hoar thickness, and specularity).

     The retrieval uncertainty is given as

$$\mathbf{S_i} = (\mathbf{S_a}^{-1} + \mathbf{K_i}^T\mathbf{S_e}^{-1}\mathbf{K_i})^{-1}, \tag{3}$$

with the a priori covariance matrix $\mathbf{S_a}$. The optimal solution and its a posteriori uncertainty are found if the condition

$$(\boldsymbol{x_i} - \boldsymbol{x_{i+1}})^T\mathbf{S_i}^{-1}(\boldsymbol{x_i} - \boldsymbol{x_{i+1}}) < \frac{N}{10} \tag{4}$$



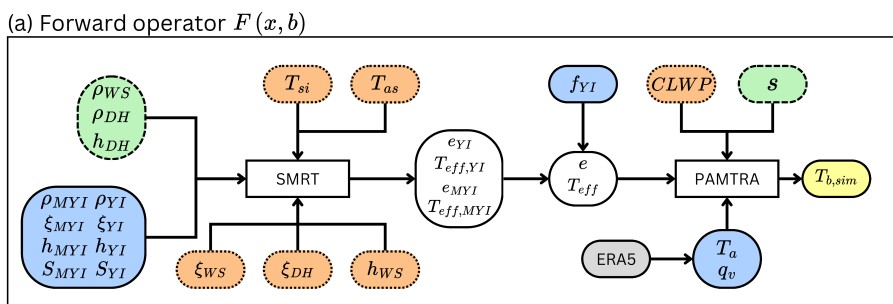

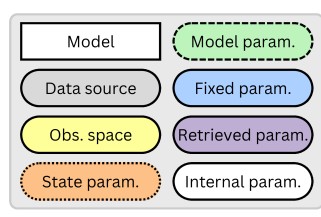

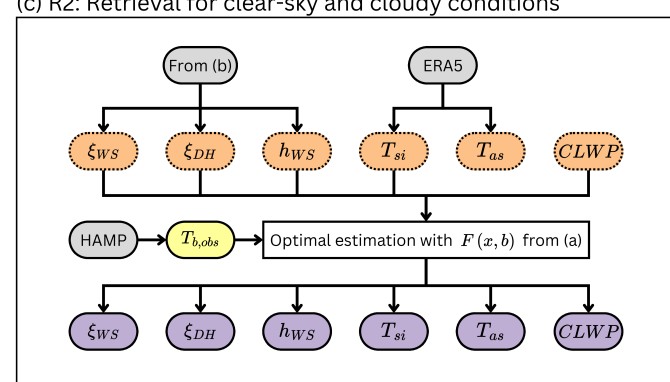

**Figure 3.** Flow diagrams of the (a) SMRT–PAMTRA forward operator ($F(x,b)$) coupled via emissivity ($e$) and effective temperature ($T_{eff}$), (b) clear-sky retrieval to calibrate snow parameters (R1), and (c) retrieval for clear-sky and cloudy conditions (R2). The parameter labeling of the forward operator in (a) corresponds to the retrieval for clear-sky and cloudy conditions in (c) (the air–snow interface temperature ($T_{as}$) and cloud liquid water path ($CLWP$) are fixed parameters in (b)). Note that the surface is characterized by fractions of young ice (YI) and multiyear ice (MYI), such that $f_{YI} + f_{MYI} = 1$. Parameter names of each symbol are listed in Table 2.

is met within six iterations. The retrieval algorithm assumes that the parameters follow a Gaussian distribution. While this is valid for most parameters, the CLWP may differ from a Gaussian distribution. A logarithmic transformation of CLWP similar to Boukabara et al. (2011) will be applied in the future.

## 3.3 Sea ice radiative transfer

The sea ice radiative transfer is solved with SMRT (Picard et al., 2018). SMRT simulates the microwave emission and scattering of horizontal and plane-parallel snow and sea ice layers. We assume that the snow consists of a mixture of ice and air without any liquid water or brine. Although liquid water likely occurred in the snowpack during parts of HALO–$(\mathcal{AC})^3$, a CLWP retrieval would be very uncertain over the highly emissive wet snow (Prigent et al., 2003; Vuyovich et al., 2017). The sea ice is characterized either as first-year or multiyear ice in SMRT. First-year ice comprises pure ice with brine inclusions, and multiyear ice comprises pure ice with brine and air inclusions. Below the sea ice, we add a semi-infinite ocean layer.



**Table 2.** State, model, and fixed parameters of the retrieval with references. The mean value of the state parameters denotes the a priori mean, while they denote the value used during the forward simulation for the model and fixed parameters. ERA5 or KT-19 mean values are derived from spatially and temporally collocated data. The standard deviation (Std.) denotes the diagonal of the a priori or model parameter covariance matrices. The minimum (Min.) and maximum (Max.) values indicate parameter limits.

| Type | Parameter | Symbol | Unit | Mean | Std. | Min. | Max. | Reference |
|------|-----------|--------|------|------|------|------|------|-----------|
| State | Cloud liquid water path | $CLWP$ | $\mathrm{g\,m^{-2}}$ | $0^b$ | 150 | 0 | 1000 | |
| | Wind slab corr. length | $\xi_{WS}$ | mm | 0.13 / $0.12^a$ | 0.05 / $0.03^a$ | 0.05 | 0.25 | Rückert et al. (2023b) |
| | Depth hoar corr. length | $\xi_{DH}$ | mm | 0.22 / $0.34^a$ | 0.1 / $0.09^a$ | 0.1 | 0.6 | Rückert et al. (2023b) |
| | Wind slab thickness | $h_{WS}$ | cm | 13 / $12^a$ | 5 / $3^a$ | 3 | 40 | |
| | Snow–ice interface temp. | $T_{si}$ | K | KT-19 / ERA5 | 3 | 243.15 | 271.15 | Schäfer et al. (2024) / Hersbach et al. (2020) |
| | Air–snow interface temp. | $T_{as}$ | K | $\text{KT-19}^b$ / ERA5 | 3 | 233.15 | 273.14 | Schäfer et al. (2024) / Hersbach et al. (2020) |
| Model | Wind slab density | $\rho_{WS}$ | $\mathrm{kg\,m^{-3}}$ | 350 | 30 | 150 | 450 | King et al. (2020) |
| | Depth hoar density | $\rho_{DH}$ | $\mathrm{kg\,m^{-3}}$ | 200 | 30 | 100 | 400 | Domine et al. (2007) |
| | Depth hoar thickness | $h_{DH}$ | cm | 15 | 3 | 3 | 40 | |
| | Specularity | $s$ | - | 0 | 0.25 | 0 | 1 | Guedj et al. (2010) |
| Fixed | Young ice fraction | $f_{YI}$ | - | 0 | | | | |
| | Air temperature | $T_a$ | K | ERA5 | | | | Hersbach et al. (2020) |
| | Specific humidity | $q_v$ | $\mathrm{kg\,kg^{-1}}$ | ERA5 | | | | Hersbach et al. (2020) |
| | Multiyear ice density | $\rho_{MYI}$ | $\mathrm{kg\,m^{-3}}$ | 850 | | | | Timco and Frederking (1996) |
| | Young ice density | $\rho_{YI}$ | $\mathrm{kg\,m^{-3}}$ | 915 | | | | Timco and Frederking (1996) |
| | Multiyear ice corr. length | $\xi_{MYI}$ | mm | 0.28 | | | | Rostosky et al. (2020) |
| | Young ice corr. length | $\xi_{YI}$ | mm | 0.15 | | | | Rostosky et al. (2020) |
| | Multiyear ice thickness | $h_{MYI}$ | cm | 150 | | | | |
| | Young ice thickness | $h_{YI}$ | cm | 30 | | | | |
| | Multiyear ice salinity | $S_{MYI}$ | psu | 1.2 | | | | Cox and Weeks (1974) |
| | Young ice salinity | $S_{YI}$ | psu | 30 | | | | |

[a] Determined from the clear-sky retrieval (Fig. 3b). [b] Fixed during the clear-sky retrieval (Fig. 3b).

The propagation and scattering of microwave radiation in sea ice and snow depend on the snow and ice microstructure (Mätzler, 2002). Here, we use the exponential autocorrelation function as microstructure representation for both snow and



sea ice, which is a function of the correlation length (Wiesmann et al., 1998). We select the improved Born approximation as electromagnetic theory to compute the scattering coefficient, which was shown to reproduce observed $T_b$ over snow from 5–243 GHz (Vargel et al., 2020; Sandells et al., 2022, 2024), and the discrete ordinate and eigenvalue radiative transfer solver (Picard et al., 2013). The permittivity of multiyear ice is calculated with the Polder–Van Santen mixing formulas. Spherical inclusions are assumed for brine in first-year ice and air bubbles in multiyear ice.

As we lack detailed sea ice and snow layer properties along the *HALO* flight track, we define two simplified sea ice types: snow-covered sea ice and bare young sea ice. The snow-covered sea ice comprises multiyear sea ice covered with a two-layer snowpack. Snow-covered first-year ice is not defined explicitly due to the limited sensitivity of frequencies above 18 GHz to the sea ice type with constant snow parameters based on SMRT (Soriot et al., 2022). The two-layer snow consists of a depth hoar and a wind slab layer, commonly observed in the Arctic (Merkouriadi et al., 2017; King et al., 2020). Wind slab typically consists of rounded snow grains, and its density is higher than the density of the underlying depth hoar. We do not retrieve the snow density, due to the limited sensitivity at the low HAMP frequencies and similar sensitivity to correlation length at high frequencies (Wivell et al., 2023). The snow thickness is set to 38 cm a priori typical for the study region in spring (Warren et al., 1999) with a depth hoar fraction of about 40 % similar to field observations (King et al., 2020). The young sea ice is simulated as bare first-year sea ice, typically present in refrozen leads that are resolved by high-resolution aircraft observations and have a higher emissivity and surface temperature than surrounding sea ice (e.g., Hewison and English, 1999; Risse et al., 2024). Note that the young sea ice fraction is included in the forward operator for sensitivity tests only and not retrieved due to poor retrieval regulation when the influence of snow parameters decreases with increasing young ice fraction and the lack of accurate a priori data under cloudy conditions. Since the sea ice type and its physical properties do not notably impact the $T_b$ at HAMP frequencies (Soriot et al., 2022), we can define a single-layer sea ice with fixed thickness, density, correlation length, and salinity (Table 2).

The sea ice and snow layer temperatures are linearly interpolated between the air–snow and snow–ice interface temperatures. An exception is the multiyear sea ice layer, where the snow–ice interface temperature is used as layer temperature, because the radiation emanates mostly from the upper part of the sea ice. Details on the a priori estimation are provided in Appendix B.

The emissivity ($e$) for each sea ice type (young ice or multiyear ice) is calculated using SMRT simulations of the upwelling brightness temperature ($T_{\mathrm{b,up}}$) with and without atmospheric downwelling brightness temperature ($T_{\mathrm{b,down}}$) following Wiesmann and Mätzler (1999), i.e.,

$$e = 1 - \frac{T_{\mathrm{b,up}}(T_{\mathrm{b,down}} = 100\,\mathrm{K}) - T_{\mathrm{b,up}}(T_{\mathrm{b,down}} = 0\,\mathrm{K})}{100\,\mathrm{K}}. \tag{5}$$

Then, the emitting layer or effective temperature ($T_{\mathrm{eff}}$) is calculated as

$$T_{\mathrm{eff}} = \frac{T_{\mathrm{b,up}}(T_{\mathrm{b,down}} = 0\,\mathrm{K})}{e}. \tag{6}$$

The emissivity and effective temperature of the two sea ice types are combined using the young ice fraction. To reduce computational cost, we simulate $e$ and $T_{eff}$ only for the center frequencies of each channel and interpolate linearly to all HAMP band passes.





## 3.4 Atmospheric radiative transfer

The atmospheric radiative transfer is simulated with PAMTRA (Mech et al., 2020). PAMTRA computes the nadir $T_b$ at the six
HAMP channels for the flight altitude of *HALO*, considering the atmospheric and surface contributions. For the surface, we provide PAMTRA with the frequency-dependent emissivity and effective temperature simulated with SMRT. The Lambertian and specular contributions to surface reflection are weighted by specularity ($s$), where $s = 0$ ($s = 1$) corresponds to a fully Lambertian (specular) surface. The specularity parameter is set to 0 as found for winter over snow (Guedj et al., 2010; Harlow and Essery, 2012), with an uncertainty accounting for 25 % specular contribution. Atmospheric profiles are used from ERA5
and not adjusted during the retrieval (Appendix A). The gas absorption model by Rosenkranz (1998) is used with modifications of the water vapor continuum absorption (Turner et al., 2009).

The a priori CLWP is set to $0\,\mathrm{g\,m^{-2}}$ with a standard deviation of $150\,\mathrm{g\,m^{-2}}$. Although CLWP is available from ERA5, we keep the retrieval simple and always assume cloud-free conditions a priori. Negative CLWP values are set to $0\,\mathrm{g\,m^{-2}}$ before calling the forward operator. The CLWP is distributed with a homogeneous cloud liquid water content between the surface and
$4\,\mathrm{km}$ height where the air temperature is above -38 °C and simulated using a monodisperse size distribution of $20\,\mathrm{\mu m}$ diameter. Both assumptions are considered to have a minimal impact on the simulated $T_b$ (Crewell et al., 2009; Ebell et al., 2017). The emission of supercooled liquid water is derived following the model by Turner et al. (2016). Rain is not included in the forward simulations because we also do not consider associated wetting of the snowpack in our SMRT setup.

Cloud ice is not included in the simulation due to the low scattering at HAMP frequencies up to 183 GHz (e.g., Buehler
et al., 2007). However, high amounts of larger snow particles lead to notable scattering from 90 to 183 GHz. For example, we observed $T_b$ depressions up to 10 K at 183±7.5 GHz during parts of the warm air intrusion over sea ice during HALO–$(\mathcal{AC})^3$. However, since we remove these cases based on the radar reflectivity threshold, we assume that the remaining snow scattering can be neglected. Adding snow water path is in principle possible, but for simplicity, we focus on the cloud liquid water signal in this work.

## 290 3.5 Synthetic retrieval setup

The synthetic retrieval allows for the quantification of the CLWP retrieval accuracy and the identification of parameter ambiguities. The observation for the synthetic retrieval consists of realistic forward simulations of a known state rather than real observations. To create realistic forward simulations that resemble natural variability, we randomly generate state and model parameters using the a priori and model parameter covariance matrices. No noise is added to the synthetic forward simulations,
but it is part of the effective measurement uncertainty of the retrieval. The synthetic database is built from random samples of HAMP observation positions and respective state, model, and fixed parameters to represent HALO–$(\mathcal{AC})^3$ conditions. The mean ERA5 integrated water vapor of the database is $5\,\mathrm{kg\,m^{-2}}$ with a standard deviation of $3\,\mathrm{kg\,m^{-2}}$, and the mean ERA5 skin temperature is -14 °C with a standard deviation of 8 K. For the CLWP accuracy assessment, we sample CLWP uniformly from 0–500 $\mathrm{g\,m^{-2}}$ and run 5000 simulations. For the identification of parameter ambiguities (2000 simulations), all parameters
are sampled from Gaussian distributions truncated by the parameter limits (Table 2).





## 4 Retrieval evaluation

### 4.1 Clear-sky evaluation

#### 4.1.1 Observation space

A comparison between the HAMP observation and simulations under clear-sky conditions (12,250 samples) provides an in-
dication of whether the SMRT–PAMTRA forward operator represents real sea ice and snow conditions. In the following, we
present $T_b$ departure statistics of the a priori and optimal states for the retrievals R1 and R2 (Fig. 4). To ensure equal sam-
pling, we analyze 81 % of the clear-sky observations where both R1 and R2 converge. Generally, R1 shows a slightly higher
convergence rate with 90 % than R2 (87 %), which is expected due to the higher number of state parameters in R2.

The departures of the optimal solution improve notably and are much narrower than the a priori for both R1 and R2,
especially from 22–118 GHz. The highest difference between the R1 and R2 distributions occurs at 90 and 118 GHz. While
R1 tends to underestimate the $T_b$, R2 slightly overestimates the $T_b$ in some cases by up to 10 K. Still, both distributions align
well with the effective measurement uncertainty despite the increase in state parameters from four to six from R1 to R2. The
highest bias in R2 occurs at 22 GHz with -3 K, but the effect on the CLWP retrieval is expected to be small. The biases of the
other channels are much smaller (-0.6–1.5 K). This indicates a substantial improvement compared to the a priori with biases
between -11 K at 22 GHz and 2 K at 118 GHz. A $T_b$ bias correction could be performed at a later stage, but is not included here.
The root mean squared error of the R2 retrieval varies between 2–4 K and lies close to the effective measurement uncertainty.
Also, the correlations between observed and simulated $T_b$ of the optimal solution are very high from 31–118 GHz with 0.9–
0.93. Hence, this clear-sky evaluation shows that the retrieval finds a state that closely matches the observations, which provides
the basis for the retrieval application to synthetic and cloudy observations.

#### 4.1.2 State space

Encouraged by the good match of the retrieval with HAMP in observation space, we now analyze the corresponding retrieved
state parameters (Fig. 5). The mean and standard deviation of the retrieved states in R1 for all clear-sky observations lie mostly
close to the a priori mean and standard deviation. The largest difference occurs for the mean depth hoar correlation length,
which increases by about $1.2\sigma$ (a priori uncertainty) from the a priori to the optimal state. This increase might be related to
snow metamorphism throughout the winter, which increased depth hoar grain size and microwave scattering in this layer. This
increase in a priori depth hoar correlation length explains the differences in the a priori $T_b$ bias between R1 and R2 (Fig. 4). The
changes in the wind slab correlation length ($-0.2\sigma$) and wind slab thickness ($0.2\sigma$) are much smaller. The retrieved variability of
the wind slab correlation length is lower than the value from the literature. For the snow–ice interface temperature, a relatively
large negative deviation can be seen. This might be related to the observed negative bias of the a priori at low frequencies in
Fig. 4 and might originate from the assumed relationship in Eq. (B1) or a misrepresentation of sea ice and snow layering. An
assessment of the spatial consistency of these parameters is presented during the retrieval application in Sect. 5.





**Figure 4.** Histograms of the $T_b$ departure between clear-sky observations and forward simulations of the a priori and optimal (opt.) states retrieved with R1 and R2. Panels show the (a) 22, (b) 31, (c) 50, (d) 90, (e) 118, and (f) 183 GHz channels. Note that only times where both retrievals converge are shown (81 %) of the data). Unc.: Effective measurement uncertainty.

The distributions of the optimal parameters from the R2 retrieval shift slightly compared to the R1 retrieval, but differences are overall small (Fig. 5). This shows that the retrieved state is not very sensitive to the a priori mean, which is important for the poorly constrained snow parameters. Compared to R1, the R2 retrieval also derives the air–snow interface temperature, as this information will not be available under cloudy conditions. The retrieved temperature centers well around the ERA5-based a priori estimate, indicating that the ERA5 skin temperature is a suitable a priori choice. The root mean squared error between the retrieved air–snow interface temperature and the skin temperature from KT-19 is 2.8 K, which is similar to the ERA5-based a priori (3.1 K; not shown). For CLWP, which is also retrieved by R2 and should ideally be zero under clear-sky identified from the radar–lidar cloud mask, the root mean squared error is $112\,\mathrm{g\,m^{-2}}$ (Fig. 5f). Generally, the state distributions are realistic despite some deviations in the snow–ice interface temperature, which affect the low-frequency HAMP channels. Thus, we



**Figure 5.** Histograms of the retrieved parameters from the retrievals R1 and R2 during clear-sky observations corresponding to Fig. 4. The a priori mean and uncertainty are shown above each panel. Panels show the (a) wind slab correlation length, (b) depth hoar correlation length, (c) wind slab thickness, (d) snow–ice interface temperature minus a priori, (e) air–snow interface temperature minus a priori (R2 only), and (f) cloud liquid water path (R2 only). Note that only times where both retrievals converge are shown (81 % of the data).

conclude that the retrieval with the SMRT–PAMTRA forward operator provides a generalized representation of the sea ice and snow layer properties for a CLWP retrieval.

## 4.2 Cloud liquid water path detectability

This section analyses the CLWP detectability of the HAMP retrieval from clear-sky observations. During clear-sky conditions, the retrieved CLWP should ideally be close to $0 \, \mathrm{g \, m^{-2}}$. Hence, we can define the CLWP detectability as the 95th percentile of





retrieved CLWP under clear-sky (Fig. 6a). For all observations, about $95\,\%$ of the retrieved CLWP are below $306\,\mathrm{g\,m^{-2}}$. We identify a distinct spatial pattern in the Central Arctic, where this detectability improves a lot down to $45\,\mathrm{g\,m^{-2}}$. This decrease with increasing distance to the ice edge is shown in Fig. 6b. Between 150–200 km, the detectability decreases from 300 to below $100\,\mathrm{g\,m^{-2}}$ and remains low for further distances to the ice edge. The infrared-based analysis by Müller et al. (2025)
shows a consistent decrease of refrozen leads with distance to the ice edge. These leads and their respective high microwave emissivity and skin temperature are correlated with false CLWP detections (Fig. 6c). The high false detection for low thin ice fraction likely corresponds to thicker young ice, potentially with a snow cover, and a skin temperature comparable to surrounding sea ice. A similar $T_b$ response between CLWP and increased bare ice fraction can also be simulated with SMRT and PAMTRA (Fig. 6d). Overall, the clear-sky retrieval evaluation shows that the retrieval detects CLWP above $50\,\mathrm{g\,m^{-2}}$ at
higher distances from the ice edge and can thus be applied to cloudy scenes.

### 4.3 Cloud liquid water path accuracy

In Sect. 4.1, we proved that the forward operator and adjustment of the state parameters closely match with clear-sky HAMP observations. In the following, we analyze the CLWP retrieval skill based on synthetic retrieval experiments (Fig. 7). Generally, the retrieval is able to reproduce the real CLWP, but with a high relative uncertainty of about $125\,\%$ for CLWP below $50\,\mathrm{g\,m^{-2}}$
and growing underestimation toward high CLWP values. The high relative uncertainty of more than $100\,\%$ for low CLWP indicates the challenge in identifying thin or low-level clouds over sea ice (Turner et al., 2007). The RMSE of the synthetic experiment for low CLWP conditions can be compared with the clear-sky retrieval (Fig. 7b). At larger distances from the sea ice edge toward the Central Arctic, the clear-sky RMSE is about $30\,\mathrm{g\,m^{-2}}$, which is comparable to the RMSE estimated from the sensitivity test for low CLWP conditions. The growing bias toward high CLWP values can likely be explained by the retrieval
starting with cloud-free conditions a priori. As the CLWP exceeds multiples of its uncertainty, a growing fraction of the cloud liquid signal influences the retrieval of snow parameters with similar Jacobians, particularly the wind slab correlation length (Appendix C).

  The uncertainty estimated from the synthetic experiments holds for all conditions that meet the forward model assumptions. Mainly, the occurrence of leads, open water, wet snow, and deviations from the simple two-layer snow assumptions increases
the CLWP uncertainty and leads to biases. However, the synthetic experiments provide the only way to assess the retrieval skill due to the lack of independent CLWP data. The good performance under clear-sky conditions provides confidence that the estimated skill closely represents real conditions. While some improvements might be expected with an improved CLWP a priori information, such as ERA5, we keep the clear-sky a priori assumption for simplicity.

  We also performed sensitivity tests with two additional dual oxygen channel pairs (51.76, 52.8, 118±4.2, and 118±2.3 GHz;
not shown). The lower surface sensitivity and differential water vapor emission signal were shown to provide additional information on precipitation, especially over land (Bauer and Mugnai, 2003; Bauer et al., 2005). However, the synthetic experiments did not yield an improvement in CLWP retrieval accuracy in relation to the additional computational cost. Furthermore, we increased the number of channels starting with the lower three channels (22–50 GHz) and found the highest improvement in

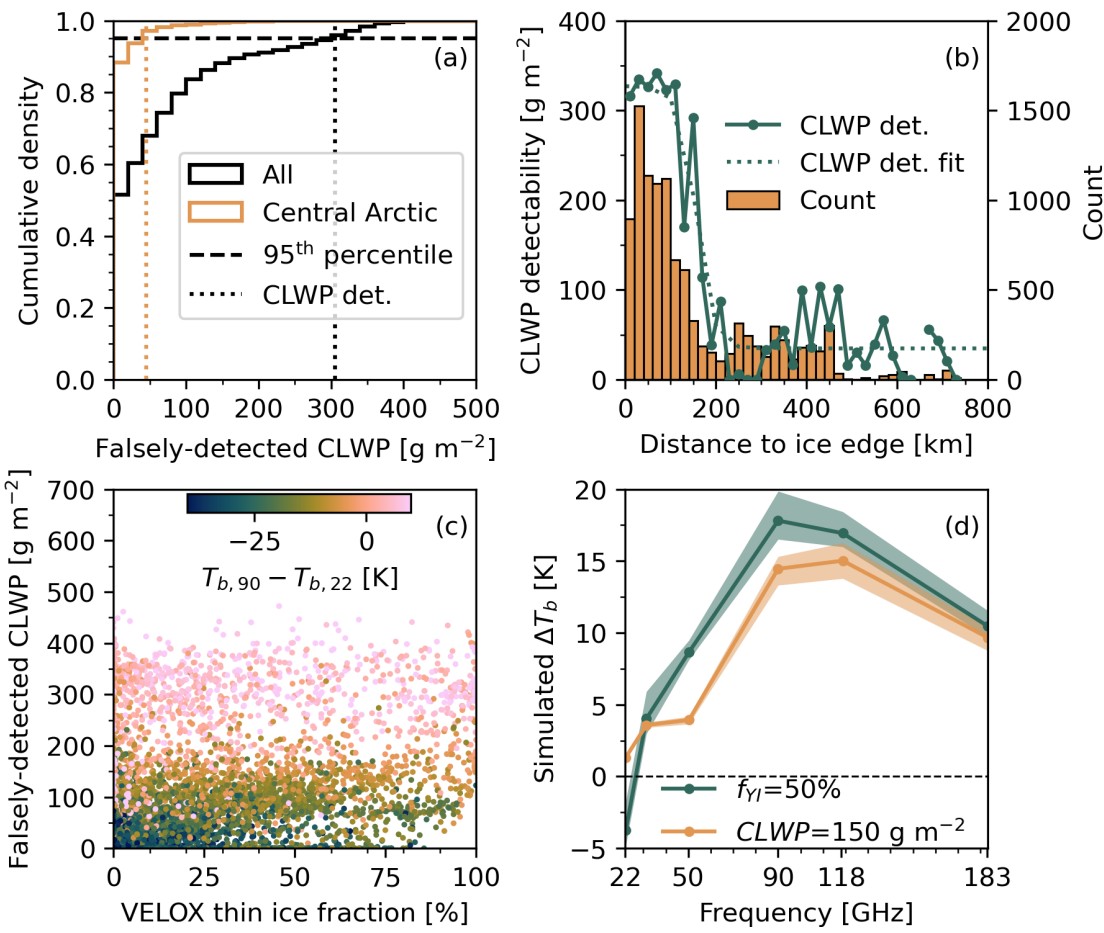

**Figure 6.** Assessment of cloud liquid water path (CLWP) detectability and falsely-detected CLWP. (a) Cumulative density of retrieved CLWP for all and Central Arctic clear-sky samples with the corresponding detectability estimated from the 95th percentile. (b) CLWP detectability and sigmoidal fit as a function of distance to sea ice edge. (c) Scatter plot between thin ice fraction derived from the thermal infrared spectral imager and falsely-detected CLWP with $T_b$ difference between 90 and 22 GHz as shading. (d) Simulated brightness temperature difference ($\Delta T_b$) between the a priori state and modified a priori states with increased $f_{YI}$ (from 0 to 50 %) and CLWP (from 0 to 150 g m$^{-2}$) for all clear-sky samples. Shading indicates the 25–75 percentile range of $\Delta T_b$ for both sensitivity tests.





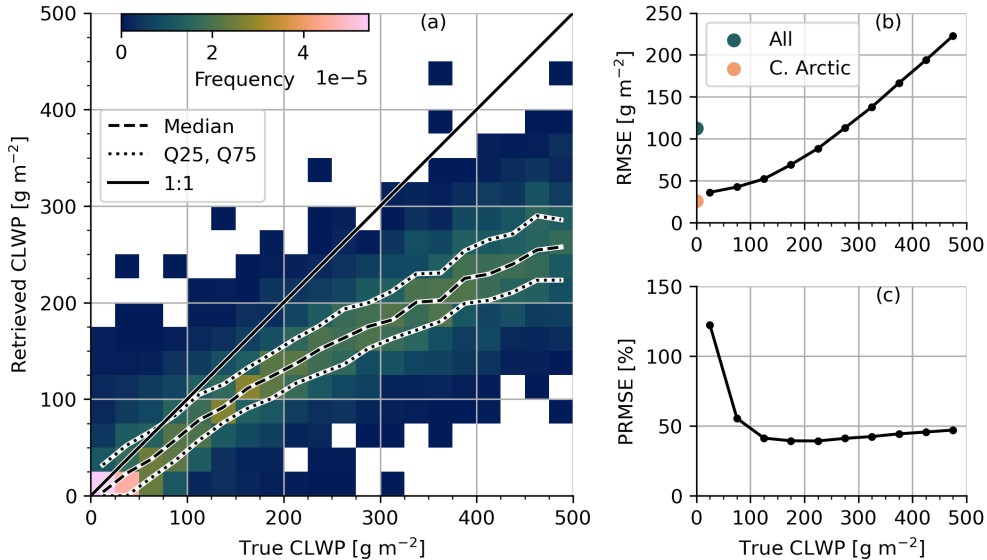

**Figure 7.** Cloud liquid water path (CLWP) retrieval skill based on synthetic experiments. (a) Joint histogram of the true CLWP used in the forward simulation and the retrieved CLWP, with median, 25th (Q25), and 75th percentiles (Q75). (b) Root mean squared error (RMSE) as a function of true CLWP from the synthetic experiments and clear-sky observations split into all and Central Arctic (C. Arctic) observations. (c) RMSE normalized by the true CLWP (PRMSE) as a function of true CLWP.

accuracy when adding the 90 GHz channel. However, we use the entire frequency range during the retrieval to provide a broad
spectral range for the surface characterization.

## 5 Retrieval application

### 5.1 Case 1: Stratocumulus (12 April 2022)

In this section, we present the HAMP retrieval for an overflight of about 800 km across a stratocumulus field over sea ice from the ice edge toward the north pole on 12 April 2022 (Fig. 8; case 1 in Fig. 1). About 92 % of the retrievals converged, which
is slightly above the convergence rate of 85 % for all flights. The near and thermal infrared images indicate refrozen leads in the initial 150 km until the stratocumulus and a cirrus layer dominate the images (Fig. 8a1–b1). The cloud top height is stable with about 300–400 m along the 250 km cross section captured by *HALO* (Fig. 8b2). The radar reflectivity signal of the cloud is rather weak with few low-reflectivity streaks (Fig. 8a2).

The observed HAMP $T_b$ generally decreases toward the north, with a breakpoint around 400 km at 22, 31, and 50 GHz
(Fig. 8c1-h1). This large-scale gradient might be related to a transition in the snow and sea ice regime toward the Central Arctic with predominantly perennial sea ice, apart from surface cooling. Small-scale features at a scale below 25 km indicate snow and sea ice variations at floe scales and the presence of refrozen leads, which likely cause the high 90, 118, and 183 GHz





$T_b$ peaks. The retrieval is able to find a matching $T_b$ for most conditions, which represents this small and large scale $T_b$ variability not represented by the a priori. An exception is the section from 450–600 km with differences of about 5–10 K, especially at 90, 118, and 183 GHz.

While no distinct cloud emission signature can be identified from the observed $T_b$ time series, the retrieval finds CLWP from 0–400 km and 650–850 km. Very high and short CLWP peaks coincide with leads due to the similar Jacobians of lead fraction and CLWP (see Fig. 6d). The broader CLWP plateau from 150–400 km might be linked to actual cloud liquid presence in the stratocumulus field. However, most CLWP values are below the CLWP detectability. The CLWP signal at the end of the segment does not align with an observed liquid layer in the lidar. The other state parameters follow the small and large scale $T_b$ features discussed earlier. Notably, the depth hoar correlation length increases around 400 km, which could be linked to more multiyear ice toward the north. Interestingly, the wind slab correlation length does not follow the same pattern and increases around 600 km.

Overall, this case study demonstrates that the retrieval finds a state space, which matches the observations under cloudy and clear-sky conditions along a long flight segment from the sea ice edge to the Central Arctic (81–88° N). However, the retrieval does not clearly identify the CLWP signature of the low-level stratocumulus field, likely due to its CLWP being below the CLWP detectability threshold, and falsely retrieves CLWP in areas without liquid cloud layers.

## 5.2 Case 2: Warm air intrusion (14 March 2022)

After analyzing the HAMP retrieval for a stratocumulus cloud with low CLWP, we now present a second case during a crossing from north to south of the warm air intrusion on 14 March 2022 (Fig. 9; case 2 in Fig. 1). Almost all of the retrievals converged along this transect (99 %). The radar shows the cloud and precipitation structure with snowfall occurring from 25–175 km. The lidar signal shows liquid top heights from 2–4 km within the precipitating system. Lower clouds occur toward the end of the segment. These clouds can also be seen in the near-infrared images. The thermal infrared images indicate increased fraction and size of leads with warmer skin temperature than the surrounding sea ice toward the end of the segment, around 550 km.

The observed HAMP $T_b$ decreases toward the end of the segment at all frequencies, which is partly linked to the decrease in atmospheric water vapor and temperature as *HALO* leaves the warm air intrusion center. This gradient is also reflected by the a priori $T_b$. However, the $T_b$ decreases well below the a priori at all frequencies with sharp boundaries at about 200 and 425 km. Similar to the stratocumulus case in Fig. 8, the observations and the simulated optimal state align well on both small and large spatial scales. Larger differences between the observations and retrieved state occur between 250–450 km. Moreover, the simulation overestimates the observed 22 GHz $T_b$ from 200 km until the end of the segment.

The retrieval adds CLWP for regions where the lidar also identifies liquid layers with up to 300 g m$^{-2}$ from 0–225 km. This region corresponds to the cloudy region at the core of the warm air intrusion and is partly excluded from the retrieval due to potential scattering by frozen hydrometeors that are not considered in the radiative transfer. The decrease in CLWP also aligns with the transition from liquid to non-liquid layers in the lidar backscatter ratio. Also, ERA5 data contains clouds with CLWP up to about 75 g m$^{-2}$, although a comparison is challenging due to the larger size of the model grid compared to the HAMP





**Figure 8.** HAMP observation and retrieval for a northbound flight segment above a stratocumulus field on 12 April 2022 (case 1 in Fig. 1). (a1) Near infrared radiance, (a2) radar reflectivity, (b1) thermal infrared $T_b$, and (b2) backscatter ratio. (c1-h1) Observation space: Observed, a priori, and optimal $T_b$ at (c1) 22, (d1) 31, (e1) 50, (f1) 90, (g1) 118, and (h1) 183 GHz. (c2-h2) State space: A priori and optimal (c2) cloud liquid water path from HAMP with ERA5 cloud liquid water path and HAMP cloud liquid water path detectability (CLWP det.), (d2) air–snow interface temperature with KT-19 skin temperature under clear-sky, (e2) snow–ice interface temperature, (f2) wind slab correlation length, (g2) depth hoar correlation length, and (h2) wind slab thickness. Note that the CLWP detectability exceeds the axis limit from 0–100 km.





footprint. The retrieved CLWP toward the end of the segment is likely associated with the low-level clouds and false detections
from refrozen leads, which formed in response to the warm air intrusion.

The air–snow interface temperature aligns well with the KT-19 skin temperature with absolute differences mostly below 2 K.
The snow–ice interface temperature drops to very low values at about 200 km, corresponding to the 22 GHz $T_b$ decrease. A
similar trend can be found for the wind slab and depth hoar correlation lengths, which increase toward the end of the segment
at about 425 km. The wind slab correlation length is very low within the precipitating system, likely due to the ambiguity with
the CLWP signal (see Appendix C). Overall, the liquid cloud signal during the warm air intrusion is well represented by the
retrieval.

## 5.3 Rain-on-snow event (12–14 March 2022)

Sea ice parameters retrieved during the warm air intrusion on 14 March 2022 partly lie outside of the expected parameter
range. This flight covered an area affected by surface melt and rain on 13 March 2022 and subsequent refreezing. It is well
known that rain-on-snow (ROS) events and associated surface glazing strongly influence the sea ice microwave signature from
ground-based (Stroeve et al., 2022) and satellite observations (Voss et al., 2003; Rückert et al., 2023a). In the following, we
present the evolution of the state parameters during three consecutive flights from 12 to 14 March 2022, covering the conditions
before, during, and after the ROS event (Fig. 10). The 91 GHz V-pol imagery captured by SSMIS onboard DMSP-F16 close
to the *HALO* overpasses shows an increase in $T_b$ by several tens of Kelvins from 12 to 13 March 2022. After the ROS event on
14 March 2022, the $T_b$ decreases far below the condition observed prior to the ROS event (Fig. 10a3), and remains low for a
couple of weeks until April 2022 (not shown).

The HAMP retrieval on 12 March lies near the a priori values and converges 87 % of the time. The only outlier is an open
water patch near the ice edge at 80° N, which corresponds to very low $T_b$ that causes artificial sharp gradients in the retrieved
snow–ice interface temperature and depth hoar correlation length. The cloudy region south of 82° N visible in radar and lidar
is captured by the retrieval. The retrieved CLWP reaches mostly values between 200 and 300 $\mathrm{g\,m^{-2}}$, which aligns with liquid
layer top heights of 4–5 km detected by the lidar.

On 13 March, a clearly visible bright band at 1 km height in the radar reflectivity profile likely indicates melting snow and
associated rainfall on the sea ice. Therefore, the retrieval is invalid for most parts of this segment and masked out by the 2 m
air temperature and radar reflectivity flags. The northern part of the bright band is not flagged at about 83° N, but the HAMP
retrieval does not converge in this area (Fig. 10a2). The area not affected by the ROS event (north of 83° N) mostly lies close
to the a priori. A notable increase in the depth hoar correlation length north of 86° N might be related to the higher fraction of
perennial sea ice.

After the ROS event, the HAMP retrieval converged for most observations (99 %) on 14 March (see Sect. 5.2). While sea ice
parameters in the northern region lie close to the a priori, they deviate from the expected range in the low-$T_b$ region south of
84.5° N (Fig. 10a3). This is slightly farther north than the observed melting layer in the radar at 83° N and could be explained
by the northward transport of warm and moist air masses between the flights and potential rain or surface melt up to 84.5° N.
Especially the wind slab correlation length and the snow–ice interface temperature lie far from the a priori and the conditions



**Figure 9.** HAMP observation and retrieval for a southbound flight segment during the warm air intrusion on 14 March 2022 (case 2 in Fig. 1). Panels as in Fig. 8.





observed before the ROS event on 12 March. Potential reasons for the altered sea ice emissivity could be the formation of
       ice lenses after the freeze-up at the surface. Ice lenses are weakly scattering and lower the microwave emissivity through the
       dielectric contrasts between adjacent layers of different densities. Additionally, newly accumulated snow on top of the ice
       lens could amplify the $T_b$ reduction. Interestingly, a secondary increase in the wind slab correlation length occurs as *HALO*
       approaches the $T_b$ minimum of the SSMIS swath around 82.5° N. Near and thermal infrared images do not show apparent
surface patterns that correlate with this microwave signature (not shown). Thus, we assume that spatial variations of snowpack
       changes (ice lens, fresh snow) contribute.

## 5.4    CLWP variability during HALO–$(\mathcal{AC})^3$

In this section, we exploit the collocated radar–lidar cloud remote sensing data from *HALO* to assess CLWP distributions for
different cloud types (Fig. 11). In total, $85\,\%$ of HAMP retrievals converge during the entire campaign, which is similar to
the clear-sky convergence rate ($87\,\%$). The CLWP distributions of HAMP shift gradually toward higher values with increasing
liquid top height determined from the lidar (Fig. 11a). The low liquid top class predominantly shows CLWP below $25\,\mathrm{g\,m^{-2}}$
and the high liquid top class shows a broad peak from $100$–$200\,\mathrm{g\,m^{-2}}$. During the absence of a liquid layer, the CLWP follows
the clear-sky distributions (Fig. 5f) with a considerable amount of falsely detected CLWP likely related to refrozen leads. To
exclude these cases, we filter for the Central Arctic where fewer leads are expected (Fig. 11c). The no liquid class remains
mostly below $50\,\mathrm{g\,m^{-2}}$, which aligns with the lower CLWP detectability threshold found in this region. Most CLWP values
above $100\,\mathrm{g\,m^{-2}}$ align with observations with liquid top heights between $1$–$5\,\mathrm{km}$. In the $50$–$100\,\mathrm{g\,m^{-2}}$ range, the $0.5$–$1\,\mathrm{km}$
liquid top heights become more frequent.

       The distributions from ERA5 follow a similar shape as the HAMP distributions for all observations (Fig. 11b) and the Central
Arctic (Fig. 11d). A notable difference for all cases is the higher number of extremes derived from HAMP, which likely relates
to the small footprint size of HAMP with $1\,\mathrm{km}$ compared to a spatial resolution of $31\,\mathrm{km}$ of ERA5. Moreover, potential false
detection over leads could cause artificial CLWP peaks. Both CLWP distributions peak at $125$–$150\,\mathrm{g\,m^{-2}}$ when a liquid layer
was detected by the lidar. For the distribution in the Central Arctic, both HAMP and ERA5 show CLWP up to $175\,\mathrm{g\,m^{-2}}$ and
a few extremes above $200\,\mathrm{g\,m^{-2}}$ mostly linked to high liquid tops.

       The analysis of CLWP distributions for HAMP and ERA5 indicates agreement in both the shape and magnitude of CLWP.
However, the relatively high uncertainty of the HAMP retrieval and the negative bias found for high CLWP from synthetic
experiments should be considered when evaluating ERA5 CLWP.

## 6    Conclusions

Passive microwave observations provide high spatial and temporal coverage in the Arctic onboard polar orbiting satellites, but
their use remains limited due to the variable sea ice and snow emission. Here, we exploited nadir-viewing passive microwave
observations ($22$–$183\,\mathrm{GHz}$) and collocated active cloud remote sensing data for diverse cloud and sea ice conditions captured
with the *HALO* aircraft during the Arctic spring HALO–$(\mathcal{AC})^3$ campaign (Wendisch et al., 2024). We developed a physical



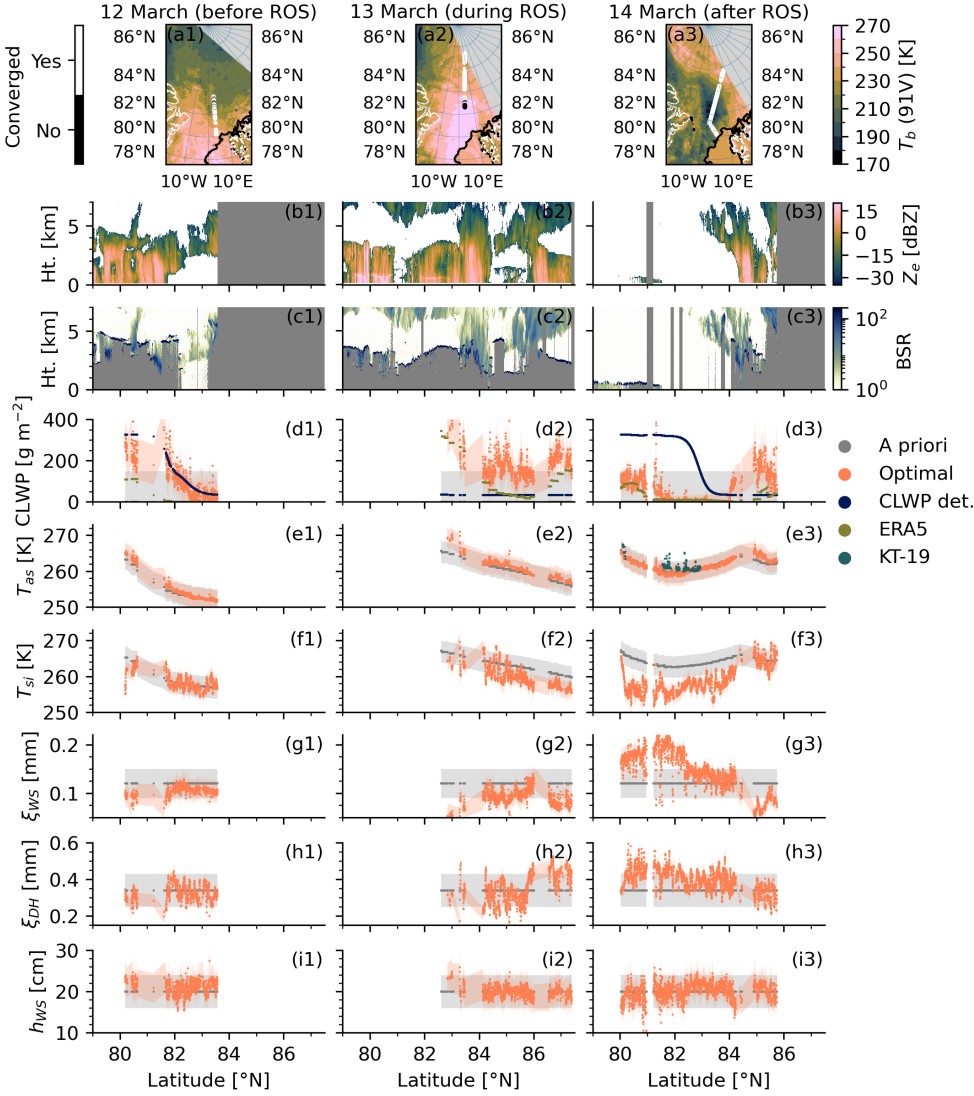

**Figure 10.** HAMP retrieval and satellite observations before the rain-on-snow (ROS) event (12 March 2022, column 1), during the ROS event (13 March 2022, column 2), and after the ROS event (14 March 2022, column 3). (a1–a3) 91 GHz V-pol $T_b$ from SSMIS onboard DMSP-F16 at about (a1) 13:30 UTC, (a2) 15:00 UTC, and (a3) 14:45 UTC, 15 % sea ice concentration contour, and meridional *HALO* flight tracks with retrieval convergence mask as shading from (a1) 13:56–15:42 UTC, (a2) 13:43–15:30 UTC, and (a3) 13:26–16:45 UTC. Note that *HALO* flew a zonal segment during the turn in (a3), not shown here. Panels below the maps show *HALO* observations and retrieval parameters (a priori and optimal) as a function of latitude: (b1–b3) Radar reflectivity, (c1–c3) backscatter ratio (BSR), (d1–d3) cloud liquid water path from HAMP with ERA5 cloud liquid water path and HAMP cloud liquid water path detectability (CLWP det.), (e1–e3) air–snow interface temperature with KT-19 skin temperature under clear-sky, (f1–f3) snow–ice interface temperature, (g1–g3) wind slab correlation length, (h1–h3) depth hoar correlation length, and (i1–i3) wind slab thickness.



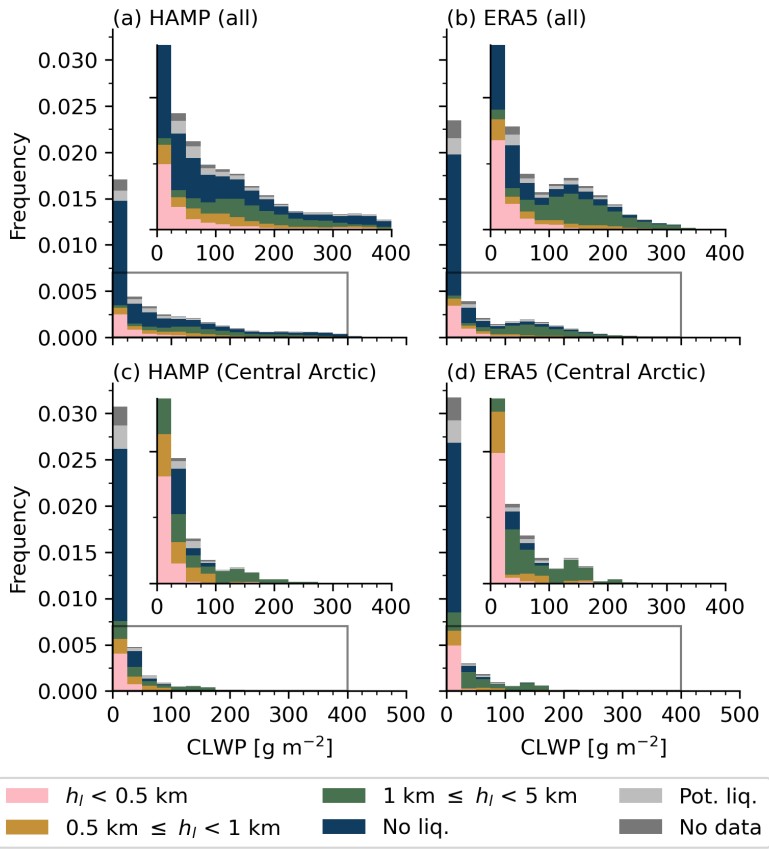

**Figure 11.** Cloud liquid water path histogram along the HALO flight track from (a, c) HAMP and (b, d) ERA5. (a, b) All observations and (c, d) observations in the Central Arctic. Shading classifies observations based on liquid top height ($h_l$), no liquid (No liq.), potential liquid (Pot. liq.), and missing lidar data. The inset region is denoted by the gray rectangle. Note that ERA5 is only shown for times where the HAMP retrieval converged.

sea ice–atmosphere optimal estimation retrieval algorithm with the loosely coupled SMRT (Picard et al., 2018) and PAMTRA (Mech et al., 2020) radiative transfer models for the HAMP microwave radiometer channels. The algorithm retrieves three snow layer parameters (wind slab correlation length, depth hoar correlation length, and wind slab thickness), the air–snow
and snow–ice interface temperatures, and CLWP. The combination of this passive microwave retrieval with *HALO's* cloud observatory instrument suite, which is typically not available for passive microwave observations from satellites, provides a unique opportunity to (1) assess the representation of sea ice and snow microwave emission by the forward model, (2) estimate the CLWP detectability and retrieval accuracy, and (3) analyze the spatial variability of CLWP over sea ice during HALO–$(\mathcal{AC})^3$.

The optimal estimation retrieval found a geophysical state consistent with HAMP clear-sky observations identified by the collocated radar–lidar cloud mask with a convergence rate of 87 %. The $T_b$ departure of the optimal solution strongly improved



compared to the a priori, which assumed no spatial variability of the three snow properties. Moreover, the distributions of all snow parameters lie within the expected range.

The CLWP detectability was assessed from the clear-sky performance. We find a detectability threshold of $50\,\mathrm{g\,m^{-2}}$ in the Central Arctic, which increases towards the marginal ice zone up to $350\,\mathrm{g\,m^{-2}}$. The detectability near the marginal ice zone can be potentially improved with additional information on lead or young ice fraction. From SMRT–PAMTRA simulations, we found that a young ice fraction of $50\,\%$ causes a similar $T_b$ signature as $150\,\mathrm{g\,m^{-2}}$ CLWP. The CLWP retrieval accuracy was derived from synthetic retrieval experiments. The relative root mean squared error of CLWP decreased from above $100\,\%$ for CLWP below $50\,\mathrm{g\,m^{-2}}$ to below $50\,\%$ above $100\,\mathrm{g\,m^{-2}}$. The identified bias for higher CLWP values can be explained by the parameter ambiguity between CLWP and wind slab correlation length.

The retrieval was applied to a stratocumulus case and a warm air intrusion case. While the CLWP of the stratocumulus case was mostly near the detectability threshold and not clearly matching with the liquid layer visible in the lidar observations, the higher CLWP of the warm air intrusion event aligned well with the lidar. The simultaneously derived surface parameters for the stratocumulus case follow realistic spatial gradients with increased snow correlation length and thus increased scattering from the ice edge toward the north pole. A statistical comparison of CLWP for all flights between HAMP and ERA5 found generally a similar CLWP shape and increasing CLWP for increasing liquid top height.

This work implies that SMRT could be useful in a coupled sea ice–atmosphere data assimilation system using a priori data from a thermodynamic sea ice and snow evolution model. This approach would benefit from the high sensitivity of passive microwave observations to snow emissivity changes due to atmospheric processes, such as rain, snowfall, and near-surface air temperature. For example, the warm air intrusion in March 2022 and the associated surface melt and rain-on-snow event altered the emissivity within a few hours due to increased liquid water fraction within the snowpack, and the subsequent refreezing and potential ice lens formation altered the $T_b$ signature for several weeks compared to conditions prior the warm air intrusion.

There are several limitations of this study, apart from the limited seasonal and spatial generalization of field observations. First, no independent quantitative reference observations of the state variables exist at scale, especially under cloudy conditions and regarding snow microstructure. The second limitation involves the simplified two-layer snowpack without fresh surface snow from accumulating snow or ice lenses from melt-freeze cycles, which both impact the emissivity at frequencies sensitive to CLWP (e.g., Sandells et al., 2024). The difficulty of simulating these scattering signatures with our two-layer snow setup might partly explain the non-convergence rate of $13\,\%$ under clear-sky conditions.

A way for future work to advance the use of passive microwave observations in the Arctic lies in exploiting the influence of atmospheric events on surface emission, such as rain-on-snow, and the integration of more observation geometries from microwave imagers and sounders. This would not only increase the temporal resolution, which is crucial during extreme events like warm air intrusions, but also ensure spectral, angular, and polarization consistency of the radiative transfer simulations. Especially the angular and polarization dependence over sea ice requires validation of SMRT, e.g., using ship observations (?Rückert et al., 2025), and could provide additional information benefit for integrated sea ice–atmosphere retrievals.



*Code and data availability.* The code for this study is available on Zenodo at https://doi.org/10.5281/zenodo.17064296 (Risse, 2025b). The optimal estimation retrieval inputs and outputs are available on Zenodo at https://doi.org/10.5281/zenodo.15848709 (Risse, 2025a). The version of PAMTRA with an emissivity vector extension corresponds to commit fb71f43 (last access: 21 November 2024), pulled from https://github.com/nrisse/pamtra/commit/fb71f43 (Mech et al., 2020). The version of SMRT corresponds to commit 6f7dadc (last access: 29 October 2024), pulled from https://github.com/smrt-model/smrt/commit/6f7dadc (Picard et al., 2018). The version of pyOptimalEstimation cor-

responds to commit 1eb4f26 (last access: 22 November 2024), pulled from https://github.com/maahn/pyOptimalEstimation/commit/1eb4f26 (Maahn et al., 2020). HAMP measurements were obtained from https://doi.org/10.1594/PANGAEA.974108 (Dorff et al., 2024), and the updated version of the bias correction used here will be made available soon. WALES measurements were obtained from https://doi.org/10.1594/PANGAEA.967086 (Wirth and Groß, 2024). KT-19 measurements were obtained from https://doi.org/10.1594/PANGAEA.967378 (Schäfer et al., 2024). VELOX measurements were obtained from https://doi.org/10.1594/PANGAEA.963382 (Schäfer et al., 2023b). Spec-

MACS measurements were obtained from https://doi.org/10.1594/PANGAEA.966992 (Weber et al., 2024b). Dropsonde measurements were obtained from https://doi.pangaea.de/10.1594/PANGAEA.968900 (George et al., 2024). The VELOX surface classification data is currently accessible upon request and will be made publicly available on PANGAEA (Müller et al., 2025). Aircraft position and orientation were obtained from the "ac3airborne" intake catalog (Mech et al., 2022). The sea ice concentration data from the University of Bremen were obtained from https://data.seaice.uni-bremen.de (last access: 30 April 2025, Spreen et al., 2008). The ERA5 reanalysis data on model levels

were obtained from https://doi.org/10.24381/cds.143582cf (last access: 28 June 2025, Hersbach et al., 2017). The ERA5 reanalysis data on single levels were obtained from https://doi.org/10.24381/cds.adbb2d47 (last access: 28 June 2025, Hersbach et al., 2023). The Level 1C $T_b$ data for SSMIS on DMSP-F16 were obtained from https://doi.org/10.5067/GPM/SSMIS/F16/1C/07 (Berg, 2021).

## Appendix A: Atmospheric profiles

The approach of fixed atmospheric temperature and specific humidity profiles differs from sea ice–atmosphere retrievals that

derive temperature profiles (Kang et al., 2023) or integrated water vapor (IWV) (Rückert et al., 2023b) based on climatological mean a priori data. The collocated instantaneous ERA5 data provide accurate IWV when compared to dropsondes launched over sea ice with a root mean squared error (RMSE) of $0.25\,\mathrm{kg\,m^{-2}}$ and percentage RMSE (PRMSE) of $9\,\%$, without notable improvement for assimilated dropsondes. Similarly low RMSE between ERA5 and dropsondes over sea ice are found for profiles of temperature ($0.5\,\mathrm{K}$ above $1\,\mathrm{km}$ height and up to $2\,\mathrm{K}$ below $1\,\mathrm{km}$ height) and relative humidity ($10\text{–}15\,\%$) (see Fig.

3 in Walbröl et al., 2024). To assess the impact of ERA5 IWV uncertainty at HAMP frequencies, we conduct a sensitivity test by increasing IWV by $10\,\%$. This test results in maximum $T_b$ changes of up to $2\,\mathrm{K}$ at $183\pm7.5\,\mathrm{GHz}$ for IWV ranges of 2 to $4\,\mathrm{kg\,m^{-2}}$ and $1.5\,\mathrm{K}$ at $118\pm8.5\,\mathrm{GHz}$ for IWV ranges of 8 to $13\,\mathrm{kg\,m^{-2}}$ (not shown). These relatively moderate sensitivities support our use of fixed atmospheric profiles.

## Appendix B: A priori interface temperatures

The snow–ice interface temperature ($T_{si}$) a priori is computed by a simple linear scaling between the air–snow interface temperature a priori ($T_{as}$) and the water temperature ($T_w = 271.35K$) with a manually determined scaling factor ($a = 0.25$),





chosen from sensitivity tests, as

$$T_{si} = T_{as} + a \cdot (T_w - T_{as}).$$ (B1)

The ERA5-based air–snow interface temperature a priori used in R2 is derived from ERA5 skin temperature ($T_{s,ERA5}$) with an empirical correction to remove biases with respect to the KT-19 skin temperature. We derive the following empirical relationship from clear-sky HALO–$(\mathcal{AC})^3$ data:

$$T_{as} = 0.94 \cdot T_{s,ERA5} + 11\,\mathrm{K}.$$ (B2)

## Appendix C: Parameter ambiguities

In the following, we analyze parameter ambiguities between the six retrieved state parameters and the four fixed model parameters from synthetic retrievals. The ambiguities are quantified from correlations between the normalized residuals derived for the state parameters as

$$r_{x,i} = \frac{x_{op,i} - x_{true,i}}{\sqrt{S_{a,ii}}},$$ (C1)

with the retrieved state $x_{op}$ and state used for the synthetic observation $x_{true}$. The same equation is adapted to the model parameters, which are fixed during the retrieval, i.e,

$$r_{b,i} = \frac{b_i - b_{true,i}}{\sqrt{S_{b,ii}}},$$ (C2)

where $b_{true}$ denotes the model parameter used for the synthetic observation. Correlations between two model parameters are neglected here as they are not directly relevant for the retrieval performance.

In total, 14 out of 39 parameter combinations show correlations larger than $\pm 0.1$, and five of the relationships include CLWP (Fig. C1). The highest correlation is found between CLWP and wind slab correlation length (0.67). This indicates that scattering in the wind slab layer partly compensates the spectral cloud emission signature and vice versa. This is consistent with similar Jacobians, which indicates that both parameters affect the simulated $T_b$ in a similar way (not shown). Also, the posterior covariance matrix shows a high correlation of 0.8 between CLWP and wind slab correlation length. Negative correlations are found between CLWP and air–snow interface temperature (-0.21) and wind slab density (-0.18). Minor relationships occur between CLWP and the specularity parameter and snow–ice interface temperature. The lower correlation with these parameters can be related to the larger differences in the Jacobian matrix. Correlations between other parameters are also found, notably between depth hoar correlation length and depth hoar thickness (-0.63) and snow–ice interface temperature (0.51). Overall, the pronounced ambiguity between CLWP and wind slab correlation length suggests that HAMP observations can only partly separate both signals over sea ice from real observations.

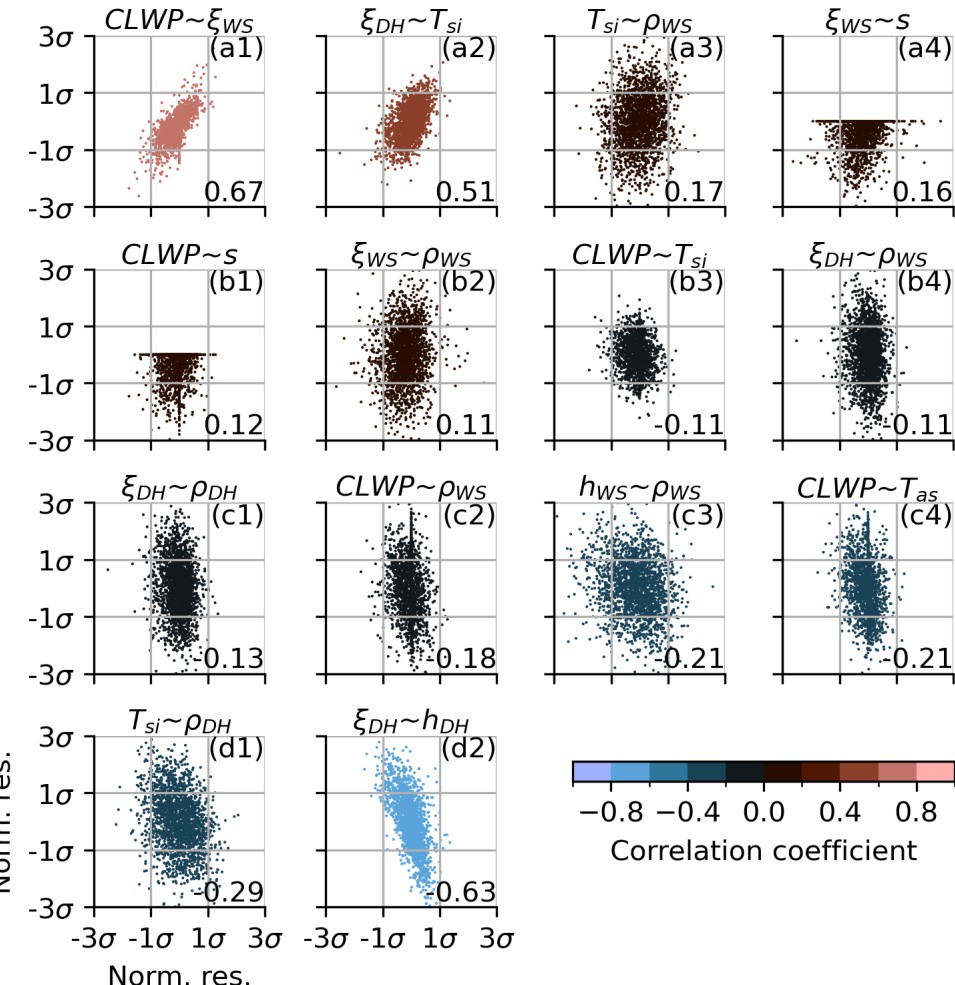

**Figure C1.** Correlations between normalized parameter residuals from the synthetic retrieval experiment. The parameter combinations are sorted from positive to negative correlations, and the first (second) parameter is shown on the horizontal (vertical) axis. Note that parameter combinations with correlations within ±0.1 are not shown and that no positive residuals occur for the specularity model parameter. Parameter names of each symbol are listed in Table 2.



*Author contributions.* NR conducted the retrieval, data analysis, and visualization, and prepared the manuscript. SC, MM, CP, and NR
conceptualized the study. SC, MM, and NR carried out the field observations. JM derived the VELOX surface classification within the
radiometer footprint. All authors reviewed and edited the manuscript.

*Competing interests.* The authors declare that they have no conflict of interest.

*Acknowledgements.* We gratefully acknowledge the funding by the German Research Foundation [Deutsche Forschungsgemeinschaft (DFG)]
of the Transregional Collaborative Research Center SFB/TRR 172 "Arctic Amplification: Climate Relevant Atmospheric and Surface Pro-
cesses, and Feedback Mechanisms $(\mathcal{AC})^3$" (Project-ID 268020496). The authors are grateful to the AWI for providing and operating the
two Polar 5 and Polar 6 aircraft. We thank the crews and the technicians of the three research aircraft for excellent technical and logistical
support. The generous funding of the flight hours for the Polar 5 and Polar 6 aircraft by AWI and for HALO by the DFG, Max-Planck-Institut
für Meteorologie (MPI-M), and Deutsches Zentrum für Luft- und Raumfahrt (DLR) is greatly appreciated. We are further thankful for the
funding provided by DFG within the framework of Priority Program (SPP 1294) to promote research with HALO (grant no. 316646266). We
would like to thank Ghislain Picard for his support on the SMRT simulation setup. Furthermore, we acknowledge the freely available Python
packages, including but not limited to numpy (Harris et al., 2020), pandas (McKinney, 2010), xarray (Hoyer and Hamman, 2017), and scipy
(Virtanen et al., 2020) for data analysis, and matplotlib (Hunter, 2007), seaborn (Waskom, 2021), and cartopy (UK Met Office, 2023) for vi-
sualization. We sincerely appreciate Fabio Crameri for providing scientific colormaps via an open repository, enhancing the visual quality of
this work (Crameri, 2018). We acknowledge the use of large language models by OpenAI via GitHub Copilot for code generation assistance
during the development of the retrieval and analysis.



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
