# Peer review of "Cloud liquid water path detectability and retrieval accuracy from airborne passive microwave observations over Arctic sea ice"

_EGUsphere, 2025_

## Referee Comment (RC3)

**Review of "Cloud liquid water path detectability and retrieval accuracy from airborne passive microwave observations over Arctic sea ice" by Nils Risse et al.**

The authors present a retrieval algorithm for cloud liquid water path (CLWP) over sea ice using passive microwave observations between 22 and 183 GHz. The method, based on optimal estimation theory, attempts to account for the highly heterogeneous surface emissivity by loosely coupling the PAMTRA atmospheric radiative transfer model with the SMRT model, which simulates the microwave radiative transfer through surface snow and ice. They evaluate the performance of the retrieval and apply the method to aircraft observations obtained during the HALO-(AC)3 field campaign.

This is a substantial piece of work and represents a significant attempt to progress our ability to exploit satellite observations in polar regions. The complex methodology is clearly described and the results are presented well. However, to me the study shows that it is, in fact, very difficult to retrieve meaningful CLWP over arctic sea ice in many circumstances because the signal due to the cloud cannot be separated from the signal due to the surface properties. This is discussed by the authors, but in my opinion it is not given sufficient prominence and the results are presented in an overly positive a manner. I therefore think that major revisions are required before the manuscript is suitable for publication.

**Major comments**

As discussed by the authors, for a successful retrieval of cloud liquid water over sea ice it is necessary for the parametrization of the surface within the model to be realistic enough such that, in the absence of cloud, the brightness temperature departures in the retrieved state are small compared to the cloud-induced signals. These departures are presented in Sec. 4.1.1, and the authors state that "the retrieval finds a state that closely matches the observations" (L319). However, Fig. 4 shows that this is not, in fact, the case. At many frequencies the R1 (clear sky) retrieval does not find a state that matches the observations to within the effective measurement uncertainty, particularly at 118 and 183 GHz, where differences of up to 10K are seen. This implies to me that the surface parametrization is not sufficiently representative, and these departures need further analysis. It may be the case that they represent the situations with significant young ice fractions, but this is not discussed in the text. As a minimum, perhaps Fig. 4 could be divided into "central arctic" and "all" regions as in the later analysis to explore this point. The clear-sky surface classification from VELOX could also be used to separate these results by the thin ice fraction. The source of any remaining discrepancies in the absence of young ice should be further discussed.

Even in the absence of significant young ice fraction, the synthetic results presented in Sec 4.3 demonstrate that the retrieval has significant problems reproducing the true CLWP. The synthetic retrieval assumes ideal circumstances where the model representation is fully accurate, but it is still not able to predict the correct CLWP. The RMSE is dominated by a significant bias, where CLWP is underestimated by almost 50% in all cases. This is not consistent with the statement that "Generally, the retrieval is able to reproduce the real CLWP" (L358). For me, this is a key finding of the study and deserves further detailed analysis. The

authors attribute the large absolute bias at higher CLWP values to the use of a cloud-free prior and that the larger CLWP values are multiples of the assumed standard deviation. This could be tested by performing synthetic retrievals with different prior values or larger standard deviations. I would also recommend testing a retrieval of log(CLWP) (as suggested in L224), as the current assumption of a strongly truncated zero-mean Gaussian distribution is not consistent with the mathematic formalism of the OEM, which assumes the retrieval parameters have a true Gaussian distribution. The ability, or otherwise, of the retrieval to distinguish between surface and cloud parameters is also important and deserves more attention and prominence. The substance of Appendix C, particularly the strong correlation between CLWP and wind slab correlation length, should be included in the main text of the paper. It would also be instructive to show the signature in brightness temperature space of each of the retrieval parameters, in a similar way to the plot in Fig. 6(d), i.e. show the change in simulated brightness temperature for a perturbation in each of the retrieval parameters by the assumed standard deviation. This will emphasize which parameters give similar brightness temperature sensitivity, and hence cannot be easily distinguished by the retrieval. The impact of these ambiguities should be discussed in detail.

The paper should also give a more balanced discussion of the retrieval performance as shown by the case studies, and the implications for our understanding of arctic clouds. For example, Case 1 appears to show that it has little value for observing low-level stratiform cloud since most of the CLWP is below the detectability threshold and there is a significant amount of false detection. It would be helpful to discuss the retrieval performance in the context of the expected climatology of liquid cloud in polar regions. The paper references existing studies using ship-based measurements (e.g. Walbröl et al., 2022) that could be used to provide this context.

**Minor comments**

L40: replace "largely" with "significantly"

L117: replace "in the lowest about 100m" with "in approximately the lowest 100m"

L119: replace "with about" with "at approximately"

L130: How could you identify scenes which might be affected by frozen hydrometeor scattering in the absence of co-located radar?

Sec 3.1: I would like a clearer explanation in this section of the difference between "state", "model" and "fixed" parameters.

Sec 3.2: Include reference for OEM method, e.g. INVERSE METHODS FOR ATMOSPHERIC SOUNDING: THEORY AND PRACTICE, Rodgers (2000) p85-86

L222: What is N? I can't see where it is defined.

L247: If I add together the prior values of wind slab and depth hoar thickness in Table 2 I get 28cm, not 38cm as stated here. Please explain this difference.

Fig 3: I found it slightly confusing that CLWP is listed as an input in R1 (clear-sky retrieval). Perhaps indicate on the figure that it is fixed at 0.

L230: Is it possible to identify the presence of wet snow from the passive microwave observations?

Table 2: Caption – Presumably the standard deviation is the square root of the diagonal of the covariance matrix, since the diagonal should be the variance.

Sec. 3.5 Synthetic retrieval setup: I would prefer to see this information in Sec 4.3 where the synthetic retrieval is discussed.

Fig 4: The legend is confusing, with the open rectangles representing the a-priori departures. Please replace these with simple lines to match what appears in the plots.

L351: I do not understand how Fig 6(c) shows a correlation between thin ice fraction and falsely detected CLWP. The figure appears to show falsely detected CLWP appearing at all values of thin ice fraction. Please explain further.

L359: A bias of 50% does not support the statement that the retrieval can reproduce the real CLWP.

L386: I cannot clearly see a cirrus layer in the radar or lidar plots in Fig. 8. Perhaps the colour scales need to be adjusted.

Fig 8: In panel (b1) the cross-track data does not add much information and makes it harder to see the magnitude of the IR temperature. Perhaps consider using a line plot of the KT-19 brightness temperature, rather than the imager data?

Fig 8 (and similar): What does the shaded orange region represent in the state space plots?

L428: Even though the air-snow interface temperature shows a similar magnitude to the KT-19 data in the clear sky region there is a different spatial trend, suggesting that the retrieval is not capturing the true behaviour.

L433: How do you know the liquid cloud signal is well represented? It's in significant disagreement with ERA5, and as mentioned is correlated with a strong decrease in wind-slab correlation length which causes strong ambiguity with the CLWP.

L480: Why not average the HAMP retrievals to a resolution of 31km to more fairly compare with ERA5?

L517-522: The study does not discuss using surface property information from a thermodynamic model, so this statement is highly speculative. The forward model used in the study does not include the effect of rain on snow, surface melt or refreezing so it is not clear how the microwave signals due to these changes could be used.

---

## Referee Comment (RC4)

Cloud water path detectability and retrieval accuracy from airborne passive microwave observations over Arctic sea ice

**General remark**

This manuscript describes an excellent experiment, possibly leading to an advancement in microwave remote sensing of the arctic ocean and atmosphere by a proper combination of forward models of microwave emission, SMRT and PAMTRA, for essential contributors, such as sea ice, snow, cloud liquid water, water vapour, and dry air. Due to the large number of physical parameters of the atmosphere and of the arctic ocean, the task required a delicate choice of observables, of variable and of fixed model parameters to get reasonable assessments. The observations consist of ERA5 data and of measurements from a large sensor package (HAMP) on a high-altitude aircraft (HALO) with microwave radiometers (and others) looking in nadir direction at frequencies between 22 and 190 GHz.

The complexity of this work required several days of reading and thinking to get a reasonable understanding. Finally, the gain of insight was great. But I might have been lost without my experience with microwave signatures of sea ice, snow and tropospheric water because basic signatures are missing in this manuscript.

**Comments, questions and corrections**

1 Change: CWP to ILW

The essential parameter is called Cloud Water Path (CWP). Unfortunately, this name is not clear, and it is misleading in three ways:

- a) The word, path, is irritating, as it may indicate which path a cloud may take on its way in the atmosphere. But this is not the case.
- b) Cloud water also consists of water vapour in the air between the cloud droplets. The mass of of cloud droplets is usually smaller than the mass of the water vapour in the cloud.
- c) Clouds may also consist of frozen water.

Since the authors understand CWP as the liquid water mass per horizontal surface area, the name should be called vertically Integrated Liquid Water (ILW) mass of clouds in the atmosphere. This corresponds well to the vertically Integrated Water Vapour (IWV) mass of the atmosphere. This quantity is correctly used in the manuscript.

2 Line 4: "the variable sea ice and snow emission and scattering signatures partly mask the cloud signal..." In my view, the opposite is true: "the sea ice and snow emission and scattering signatures are partly masked by the atmosphere".

3 Line 53: What do you mean with "spatially resolved latent space representation of the sea ice"?

4 Figure 6d Simulated TB response: Shown are spectra of undefinded quantities "DeltaTB". Please correct to TB values or define DeltaTB.

5 Line 418: "Crossing of the warm air intrusion from north to south": Correct to "crossing from north to south of a warm air intrusion" (from south)

6 Line 434: What do you mean with "artificial signals"

7 Line 439: To avoid misunderstandings, connect tails of linked adjectives in front of a substantive by hyphens, such as "very low wind-slab-correlation lengths". Also elswhere.

8 Line 440: Correct or clarify (Fig. C1)

9 Appendix A: This is very limited information on atmospheric profiles. The limitation to ERA5 data is questionable, here. Warm and humid air inflows into the arctic area can change IWV by an order of magnitude within short time. Therefore, the example with 10% change of a rather dry troposphere is insufficient. Actual water vapour variations could be accounted for, e.g by pattern differences and/or short-time variations at and between 22 and 31 GHz. Also, what I found from ground-based observations is that under advective conditions, the temporal decrease of IWV often corresponds to the precipitation in between.

10 Comment on the specularity parameter *s*: This parameter is most important at nadir view direction because of the largest difference between lambert scattering and specular reflection at a horizontal surface. On the other hand, there is no polarisation information in this case. Therefore, we need off-nadir observations, too, for real tests of the specularity.

I am open for direct discussion if needed to better interact.

Christian Mätzler matzler@gamma-rs.ch

---

## Author Comment (AC1)

**Authors' Response to Reviews of**

**Cloud liquid water path detectability and retrieval accuracy from airborne passive microwave observations over Arctic sea ice**

Nils Risse, Mario Mech, Catherine Prigent, Joshua J. Müller, and Susanne Crewell
*Atmospheric Measurement Techniques,*
* * *
**RC:** *Reviewers' Comment*,     AR: Authors' Response,     ☐ Manuscript Text

**1. RC2**

**1.1. General remark**

**RC:** *The authors present a retrieval algorithm for cloud liquid water path (CLWP) over sea ice using passive microwave observations between 22 and 183 GHz. The method, based on optimal estimation theory, attempts to account for the highly heterogeneous surface emissivity by loosely coupling the PAMTRA atmospheric radiative transfer model with the SMRT model, which simulates the microwave radiative transfer through surface snow and ice. They evaluate the performance of the retrieval and apply the method to aircraft observations obtained during the HALO-(AC)[3] field campaign. This is a substantial piece of work and represents a significant attempt to progress our ability to exploit satellite observations in polar regions. The complex methodology is clearly described and the results are presented well. However, to me the study shows that it is, in fact, very difficult to retrieve meaningful CLWP over arctic sea ice in many circumstances because the signal due to the cloud cannot be separated from the signal due to the surface properties. This is discussed by the authors, but in my opinion it is not given sufficient prominence and the results are presented in an overly positive a manner. I therefore think that major revisions are required before the manuscript is suitable for publication.*

**AR:** The authors would like to thank the reviewer for providing highly valuable and constructive feedback on this manuscript. We have carefully considered all the comments and provided responses below.

**1.2. Major comments**

**RC:** *As discussed by the authors, for a successful retrieval of cloud liquid water over sea ice it is necessary for the parametrization of the surface within the model to be realistic enough such that, in the absence of cloud, the brightness temperature departures in the retrieved state are small compared to the cloud-induced signals. These departures are presented in Sec. 4.1.1, and the authors state that "the retrieval finds a state that closely matches the observations" (L319). However, Fig. 4 shows that this is not, in fact, the case. At many frequencies, the R1 (clear sky) retrieval does not find a state that matches the observations to within the effective measurement uncertainty, particularly at 118 and 183 GHz, where differences of up to 10K are seen. This implies to me that the surface parametrization is not sufficiently representative, and these departures need further analysis. It may be the case that they represent the situations with significant young ice fractions, but this is not discussed in the text. As a minimum, perhaps Fig. 4 could be divided into "central arctic" and "all" regions as in the later analysis to explore this point. The clear-sky surface classification from VELOX could also be used to separate these results by the thin ice fraction. The source*

*of any remaining discrepancies in the absence of young ice should be further discussed.*

AR: We agree that we need to be careful when evaluating the forward model performance based on brightness temperature departures under clear-sky conditions. To better illustrate the forward model performance, we present the $T_b$ departure distribution of retrieval R1 (clear-sky) in the Central Arctic (29% of the clear-sky samples) and over young sea ice, defined as >25% VELOX thin ice fraction (20% of the clear-sky samples, Fig. Rev1). The fraction of thin ice in the Central Arctic is much lower than for all data (1%; we have added this information to Sect. 2.5 and 4.2). Young ice leads to positive departures at 31–183 GHz, and negative departures at 22 GHz. This is consistent with the simulated TB response of young ice in Fig. 6d of the preprint, which shows a negative $T_b$ response for a young ice fraction of 50% at 22 GHz, while all other channels have a positive TB response. However, the $T_b$ residual at high frequencies is not as strong as in the simulations due to increments in the wind slab correlation length and thickness, which are weakly correlated with young ice fraction by -0.15 and 0.11, respectively. Both changes lead to an increase in the $T_b$ at 90 and 118 GHz, as shown in Fig. Rev5. While for R1, most observations are outside of the measurement uncertainty, the R2 retrieval improves the fit at 50–183 GHz by falsely adding CLWP (Fig. 6c of the preprint). It should also be noted that thin/young ice with a similar skin temperature to the surrounding sea ice cannot be detected by VELOX (see response to comment on L351).

In the Central Arctic, the departures of R1 are almost fully within the effective measurement uncertainty for all channels for R1. For R2, there are several cases with negative departures at 90 and 118 GHz as shown in Fig. 4 of the preprint. A direct comparison with equivalents from R1 for these cases demonstrated that R2 reduced the wind slab correlation length compared to R1 (not shown). A similar effect can also be seen from the synthetic retrieval experiment for R2 (Fig. Rev2). These are likely related to the optimization method itself rather than systematic deficiencies in the SMRT–PAMTRA forward operator. As these residuals are typically smaller than the 1-sigma CLWP signal (Fig. 6d of the preprint), we assume that they do not significantly affect the retrieval performance. We added the results of this comparison to Sect. 4.1.1.

To better understand remaining residuals of R1, we compared inter-channel correlations between observations and simulations for all and Central Arctic clear-sky cases (Fig. Rev3). We added this figure and the description to the Appendix, and included it in the discussion in Sect. 4.1.1.

RC: *Even in the absence of significant young ice fraction, the synthetic results presented in Sec 4.3 demonstrate that the retrieval has significant problems reproducing the true CLWP. The synthetic retrieval assumes ideal circumstances where the model representation is fully accurate, but it is still not able to predict the correct CLWP. The RMSE is dominated by a significant bias, where CLWP is underestimated by almost 50% in all cases. This is not consistent with the statement that "Generally, the retrieval is able to reproduce the real CLWP" (L358). For me, this is a key finding of the study and deserves further detailed analysis. The authors attribute the large absolute bias at higher CLWP values to the use of a cloud-free prior and that the larger CLWP values are multiples of the assumed standard deviation. This could be tested by performing synthetic retrievals with different prior values or larger standard deviations.*

AR: To test the role of the a priori assumptions, we conducted a synthetic retrieval experiment using uniformly varied ($0$–$500\,\mathrm{g\,m^{-2}}$) CLWP a priori mean and standard deviation. Figure Rev4 shows the true vs a priori and true vs retrieved CLWP for 1000 synthetic retrievals. Shading indicates the distance of the a priori from the true CLWP in multiples of the a priori standard deviation. Note that the synthetic experiment in the preprint Fig. 7a uses an a priori mean of $0\,\mathrm{g\,m^{-2}}$ with standard deviation of $150\,\mathrm{g\,m^{-2}}$. In the setup used in the manuscript, a true CLWP of 150, 300, and $450\,\mathrm{g\,m^{-2}}$ corresponds to -1, -2, and -3 standard deviations, respectively. The synthetic experiment demonstrates that a priori assumptions significantly impact the retrieval performance, and the median in the manuscript (Fig. 7) roughly matches the corresponding

[Figure]

Figure Rev1: Histograms of the $T_b$ departure between clear-sky observations and forward simulations of the optimal (opt.) state retrieved with R1. Panels show the (a) 22, (b) 31, (c) 50, (d) 90, (e) 118, and (f) 183 GHz channels. Note that only times where R1 and R2 retrievals converge are shown (81 % of the data). Unc.: Effective measurement uncertainty. Thin sea ice is defined as VELOX thin ice fraction above 25%.

[Figure]

Figure Rev2: Same as Fig. 4 in the preprint, but for the synthetic retrieval for the ambiguity test.

[Figure]

Figure Rev3: Inter-channel $T_b$ correlations from (a, d) HAMP observations, (b, e) optimal state simulations from R1, and (c, f) differences between the correlation from observed and optimal state simulations from R1 for (a–c) all and (d–f) Central Arctic observations.

shading in Fig. Rev4. The experiment confirms the previous assumption that there is an underestimation of CLWP when the a priori is too low. Hence, the underestimation of high CLWP values in Fig. 7a of the preprint is related to the a priori choice. However, there seems to be a slight underestimation of CLWP in this synthetic experiment for high CLWP, which could be related to the non-linear TB response of CLWP with increasing CLWP. We added a paragraph in Sect. 4.3 with the discussion from this experiment.

RC: *I would also recommend testing a retrieval of log(CLWP) (as suggested in L224), as the current assumption of a strongly truncated zero-mean Gaussian distribution is not consistent with the mathematic formalism of the OEM, which assumes the retrieval parameters have a true Gaussian distribution.*

AR: We tested the logarithmic transformation of CLWP during earlier versions of the retrieval, but a more stable retrieval performance could be achieved using a linear CLWP with a truncation around $0\,\mathrm{g\,m^{-2}}$. At this point, the technical capability to switch between the two options is not available. We believe that the current approach is sufficient for the scope of this work, and the theoretical retrieval performance is sufficiently well-characterized through the use of synthetic retrieval experiments presented in the manuscript and in Fig. Rev4.

RC: *The ability, or otherwise, of the retrieval to distinguish between surface and cloud parameters is also important and deserves more attention and prominence. The substance of Appendix C, particularly the strong correlation between CLWP and wind slab correlation length, should be included in the main text of the paper.*

AR: A discussion of the key results of Appendix C is now included in Sect. 4.3, where the synthetic retrieval experiment for CLWP accuracy estimation is analyzed.

[Figure]

Figure Rev4: Synthetic retrieval experiment with randomly varied cloud liquid water path (CLWP) a priori mean and uncertainty. (a) Distribution of randomly drawn true and a priori CLWP. (b) Distribution of retrieved CLWP as a function of true CLWP. Shading indicates the difference between a priori CLWP and true CLWP normalized by the a priori uncertainty.

**RC:** *It would also be instructive to show the signature in brightness temperature space of each of the retrieval parameters, in a similar way to the plot in Fig. 6(d), i.e. show the change in simulated brightness temperature for a perturbation in each of the retrieval parameters by the assumed standard deviation. This will emphasize which parameters give similar brightness temperature sensitivity, and hence cannot be easily distinguished by the retrieval. The impact of these ambiguities should be discussed in detail.*

AR: We added a figure of the TB change for a perturbation of each state and model parameter by one standard deviation to Sect. 3.1 (see Fig. Rev5). The results are similar to those found in the ambiguity test and Fig. 6d. We provide a description of the basic radiometric signatures before presenting any retrieval results in Sect. 4. The discussion on ambiguities is added to Sect. 4.3, using also results from the synthetic retrieval experiment in Appendix C.

**RC:** *The paper should also give a more balanced discussion of the retrieval performance as shown by the case studies, and the implications for our understanding of arctic clouds. For example, Case 1 appears to show that it has little value for observing low-level stratiform cloud since most of the CLWP is below the detectability threshold and there is a significant amount of false detection. It would be helpful to discuss the retrieval performance in the context of the expected climatology of liquid cloud in polar regions. The paper references existing studies using ship-based measurements (e.g. Walbröl et al., 2022) that could be used to provide this context.*

AR: We have revised the implications in the conclusion section to synthesize the results and link to the 1-year CLWP climatology from MOSAiC (Walbröl et al, 2022, Fig. 3).

**1.3.   Minor comments**

**RC:** *L40: replace "largely" with "significantly"*

[Figure]

Figure Rev5: Simulated sensitivity of HAMP channels to variations in the state and model parameters during HALO–$(\mathcal{AC})^3$. The sensitivity is scaled by the standard deviation of each parameter assumed during the retrieval R2.

**AR:** Done.

**RC:** *L117: replace "in the lowest about 100m" with "in approximately the lowest 100m"*

**AR:** Done.

**RC:** *L119: replace "with about" with "at approximately"*

**AR:** We slightly modified the sentence and replaced "with about".

**RC:** *L130: How could you identify scenes which might be affected by frozen hydrometeor scattering in the absence of co-located radar?*

**AR:** The identification of frozen hydrometeors from HAMP observations over sea ice could be performed in a similar way as for liquid clouds in this work by including, e.g., frozen water path into the state vector. However, compared to liquid cloud emission, the scattering signature of frozen hydrometeors strongly depends on the ice particle shape, size, orientation, and habit, which leads to uncertainties in the radiative transfer simulations (e.g., Kaur et al., 2022). In the case of mixed-phase clouds, the vertical positioning of liquid and frozen hydrometeors must also be considered (e.g., Xie et al., 2015). At HAMP frequencies, the $183\pm7.5$ GHz channel is most sensitive to the scattering by frozen hydrometeors and $T_b$ depressions up to 10 K were found under humid conditions without a significant surface contribution over sea ice. Detecting large frozen hydrometeors near the surface would need an accurate representation of the surface radiative transfer, as demonstrated for liquid clouds in this work.

Kaur, I., Eriksson, P., Barlakas, V., Pfreundschuh, S., & Fox, S. (2022). Fast Radiative Transfer Approximating Ice Hydrometeor Orientation and Its Implication on IWP Retrievals. Remote Sensing, 14(7), 1594. https://doi.org/10.3390/rs14071594

Xie, X., S. Crewell, U. Löhnert, C. Simmer, and J. Miao (2015), Polarization signatures and brightness temperatures caused by horizontally oriented snow particles at microwave bands: Effects of atmospheric absorption. J. Geophys. Res. Atmos., 120, 6145–6160. doi: 10.1002/2015JD023158.

**RC:** *Sec 3.1: I would like a clearer explanation in this section of the difference between "state", "model" and "fixed" parameters.*

AR: We have added a paragraph that describes the difference between the three parameter groups. We have also renamed 'Retrieved parameters' to 'Optimized state parameters' in the legend of Fig. 3 to highlight that these are the same set as the state parameters.

**RC:** *Sec 3.2: Include reference for OEM method, e.g. INVERSE METHODS FOR ATMOSPHERIC SOUND-ING: THEORY AND PRACTICE, Rodgers (2000) p85-86*

AR: Done.

**RC:** *L222: What is N? I can't see where it is defined.*

AR: N is the number of state parameters. We added this missing description to the text.

**RC:** *L247: If I add together the prior values of wind slab and depth hoar thickness in Table 2 I get 28cm, not 38cm as stated here. Please explain this difference.*

AR: The values for the wind slab thickness provided in Table 2 and the total snow depth in the text are not correct. The a priori mean wind slab thickness is $20\,\mathrm{cm}$ for both R1 and R2, and the uncertainty is $5\,\mathrm{cm}$ for R1 and $4\,\mathrm{cm}$ for R2. We corrected the table and the text accordingly.

**RC:** *Fig 3: I found it slightly confusing that CLWP is listed as an input in R1 (clear-sky retrieval). Perhaps indicate on the figure that it is fixed at 0.*

AR: We added a gray box with $0\,\mathrm{g}\,\mathrm{m}^{-2}$ to highlight the fixed value of CLWP in R1. For consistency, the same box is also added to R2, where $0\,\mathrm{g}\,\mathrm{m}^{-2}$ is the a priori value. The different color and outline, as well as the absence of CLWP in the group of optimized state parameters in R1, now highlight that CLWP is fixed in R1. For consistency, we also renamed the 'Data source' to 'Input' in the legend.

**RC:** *L230: Is it possible to identify the presence of wet snow from the passive microwave observations?*

AR: The presence of wet snow can be detected from satellites to derive the melt season length (e.g., Markus et al., 2009). We did not test whether a similar approach is applicable to HAMP. However, we first apply the retrieval to all observations over sea ice before screening for potentially wet snow using ERA5. We found that the retrieval converges much less often (well below 50 %) in the presence of near-melting ERA5 skin temperatures and rain in the radar (bright band). As the convergence rate is much lower than for other cases and widely spread across the affected region, we assume that this would be a good indicator for wet snow detection. Potentially, this radiative transfer-based detection approach would be superior to simple and fast satellite algorithms (e.g., gradient and polarization ratios), as it accounts for the atmospheric state and surface temperature using reanalysis. However, to avoid screening for non-melting cases, we use independent ERA5 data to exclude wet snow.

Markus, T., J. C. Stroeve, and J. Miller (2009), Recent changes in Arctic sea ice melt onset, freezeup, and melt season length, J. Geophys. Res., 114, C12024, doi:10.1029/2009JC005436.

**RC:** *Table 2: Caption – Presumably the standard deviation is the square root of the diagonal of the covariance matrix, since the diagonal should be the variance.*

AR: Yes, we updated the table caption.

RC: ***Sec. 3.5 Synthetic retrieval setup: I would prefer to see this information in Sec 4.3 where the synthetic retrieval is discussed.***

AR: We decided to keep the description of the synthetic retrieval in Sect. 3.5 as it describes the methodology not only for Sect. 4.3, but also for Appendix C (Appendix D in the revision).

RC: ***Fig 4: The legend is confusing, with the open rectangles representing the a-priori departures. Please replace these with simple lines to match what appears in the plots.***

AR: We have updated the legend symbols where open rectangles were used for lines (Fig. 4 and Fig. 6a).

RC: ***L351: I do not understand how Fig 6(c) shows a correlation between thin ice fraction and falsely detected CLWP. The figure appears to show falsely detected CLWP appearing at all values of thin ice fraction. Please explain further.***

AR: There are high falsely detected CLWP values at all values of thin ice fraction, but near $0\,\mathrm{g\,m^{-2}}$ falsely-detected CLWP occurs only for low thin ice fraction. Therefore, we find a correlation of 0.51 between VELOX thin ice fraction and falsely-detected CLWP, as shown in Fig. 6c. The reason for falsely-detected CLWP occurring at all values of the thin ice fraction is that the infrared thin ice detection is solely based on the skin temperature. Therefore, snow-covered young ice with a colder skin temperature might be missed, while a thin snow cover likely only has a small effect on the sea ice emissivity at HAMP frequencies. We added the correlation coefficient and refined the explanation in Sect. 4.2.

RC: ***L359: A bias of 50% does not support the statement that the retrieval can reproduce the real CLWP.***

AR: We have replaced "is able to reproduce" by "approximates". See also our response to the second major comment.

RC: ***L386: I cannot clearly see a cirrus layer in the radar or lidar plots in Fig. 8. Perhaps the colour scales need to be adjusted.***

AR: The cirrus layer is located at heights between $4$–$8\,\mathrm{km}$ and is not visible in the plot, which zooms on the low-level stratocumulus. We added this to the text.

RC: ***Fig 8: In panel (b1) the cross-track data does not add much information and makes it harder to see the magnitude of the IR temperature. Perhaps consider using a line plot of the KT-19 brightness temperature, rather than the imager data?***

AR: We decided to show the VELOX thermal infrared data to indicate the presence of leads during the first about 100 km of the segment. The KT-19 data is shown in panel (d2). Due to cloud presence at low altitudes and the cirrus above (not visible), only a few clear-sky KT-19 air-snow interface temperature observations occur. These are shown as green dots around 100 km. During the rest of the segment, the KT-19 data likely contains contamination by clouds and is therefore filtered using the radar and lidar cloud masks.

RC: ***Fig 8 (and similar): What does the shaded orange region represent in the state space plots?***

AR: The shaded orange region in the state space plots in Fig. 8–10 corresponds to the square root of the main diagonal of the retrieval uncertainty $\mathbf{S}_i$ in Eq. (3) after convergence is reached. The gray shading corresponds to the square root of the main diagonal of the a priori uncertainty $\mathbf{S}_a$. Similar quantities are also shown as shading in the observation space plots from the measurement uncertainty $\mathbf{S}_y$ (Obs.) and the effective measurement uncertainty $\mathbf{S}_e$ (A priori, optimal). We added this information to the captions in Figs. 8 and 10.

**RC:** *L428: Even though the air-snow interface temperature shows a similar magnitude to the KT-19 data in the clear sky region there is a different spatial trend, suggesting that the retrieval is not capturing the true behaviour.*

**AR:** We added a description of this pattern to Sect. 5.2. Further details are discussed in Sect. 5.3, where the retrieval performance is related to the rain-on-snow event and subsequent refreezing of the snow. The area with divergence of KT-19 and air–snow interface corresponds to the $T_b$ minimum of the SSMIS swath around 82.5° N.

**RC:** *L433: How do you know the liquid cloud signal is well represented? It's in significant disagreement with ERA5, and as mentioned is correlated with a strong decrease in wind-slab correlation length which causes strong ambiguity with the CLWP.*

**AR:** The retrieved CLWP in Fig. 9c2 aligns with the liquid layers visible in the lidar backscatter ratio in Fig. 9b2. We changed "well represented" to "the occurrence of liquid clouds during the warm air intrusion is well captured by the retrieval". This acknowledges that we cannot evaluate the quality of the retrieved CLWP magnitude. Regarding the comparison with ERA5, we added the correlation with HAMP for the Central Arctic (0.79) and all observations (0.51) to Sect. 5.4 (see revised methodology in our answer to the next comment on L480).

**RC:** *L480: Why not average the HAMP retrievals to a resolution of 31km to more fairly compare with ERA5?*

**AR:** We have revised the figure using hourly averages of both ERA5 and HAMP onto a 31 km equal-area grid. Moreover, we provide correlations for all observations and the Central Arctic in the text. The resampling is performed after extracting the spatially nearest ERA5 grid cell to the HALO flight track. The categories correspond to the most frequent category within the grid cell. During the revision, we noticed that panels (b) and (c) were swapped in the preprint, which has now been corrected. The ERA5 CLWP remains largely unchanged, as the resampled resolution is approximately equal to the grid size of the reanalysis. The HAMP observations show less extreme CLWP than in the previous version.

**RC:** *L517-522: The study does not discuss using surface property information from a thermodynamic model, so this statement is highly speculative. The forward model used in the study does not include the effect of rain on snow, surface melt or refreezing so it is not clear how the microwave signals due to these changes could be used.*

**AR:** While we do not use surface properties from a thermodynamic model, the experiments help to identify which processes should be captured by the thermodynamic model to improve the radiative transfer for CLWP retrievals. We have modified the paragraph on the implications. We also included a sentence in the final paragraph on future work, suggesting the replacement of the static a priori sea ice and snow information used here with output from a thermodynamic model. This would immediately enable the forward model to include rain on snow, surface melt, or refreezing, as SMRT contains dielectric permittivity parameterizations for wet snow and allows defining layers representing ice lenses that often occur after refreezing. However, it would require a substantial amount of work to test such a framework, and our static approach could serve as a baseline.

---

## Author Comment (AC2)

**Authors' Response to Reviews of**

**Cloud liquid water path detectability and retrieval accuracy from airborne passive microwave observations over Arctic sea ice**

Nils Risse, Mario Mech, Catherine Prigent, Joshua J. Müller, and Susanne Crewell
*Atmospheric Measurement Techniques,*
* * *
**RC:** *Reviewers' Comment*,     AR: Authors' Response,     ☐ Manuscript Text

**1. RC1, Prof. Dr. Christian Mätzler**

**1.1. General remark**

**RC:** *This manuscript describes an excellent experiment, possibly leading to an advancement in microwave remote sensing of the arctic ocean and atmosphere by a proper combination of forward models of microwave emission, SMRT and PAMTRA, for essential contributors, such as sea ice, snow, cloud liquid water, water vapour, and dry air. Due to the large number of physical parameters of the atmosphere and of the arctic ocean, the task required a delicate choice of observables, of variable and of fixed model parameters to get reasonable assessments. The observations consist of ERA5 data and of measurements from a large sensor package (HAMP) on a high-altitude aircraft (HALO) with microwave radiometers (and others) looking in nadir direction at frequencies between 22 and 190 GHz. The complexity of this work required several days of reading and thinking to get a reasonable understanding. Finally, the gain of insight was great. But I might have been lost without my experience with microwave signatures of sea ice, snow and tropospheric water because basic signatures are missing in this manuscript.*

**AR:** The authors would like to thank Prof. Dr. Christian Mätzler for providing highly valuable and constructive feedback on this manuscript. We have carefully considered all the comments and provided responses below.

We have added a figure to the manuscript showing the basic simulated TB signatures of model and state parameters at HAMP frequencies. See also our response to the corresponding major comment by the second reviewer.

**1.2. Comments, questions and corrections**

**RC:** *1 Change: CWP to ILW. The essential parameter is called Cloud Water Path (CWP). Unfortunately, this name is not clear, and it is misleading in three ways: a) The word, path, is irritating, as it may indicate which path a cloud may take on its way in the atmosphere. But this is not the case. b) Cloud water also consists of water vapour in the air between the cloud droplets. The mass of of cloud droplets is usually smaller than the mass of the water vapour in the cloud. c) Clouds may also consist of frozen water. Since the authors understand CWP as the liquid water mass per horizontal surface area, the name should be called vertically Integrated Liquid Water (ILW) mass of clouds in the atmosphere. This corresponds well to the vertically Integrated Water Vapour (IWV) mass of the atmosphere. This quantity is correctly used in the manuscript.*

**AR:** We renamed "cloud water path" to "cloud liquid water path (CLWP)" in the text and figures. This avoids

confusion with water vapor and ice, and aligns with the naming convention for Essential Climate Variables (ECV) by the Global Climate Observing System (GCOS).

RC: *2 Line 4: "the variable sea ice and snow emission and scattering signatures partly mask the cloud signal..." In my view, the opposite is true: "the sea ice and snow emission and scattering signatures are partly masked by the atmosphere".*

AR: We changed this to: "Spaceborne microwave radiometers provide a high sensitivity to CLWP at pan-Arctic scales, but extracting this information over sea ice requires separation of surface and cloud emission."

RC: *3 Line 53: What do you mean with "spatially resolved latent space representation of the sea ice"?*

AR: In general, the latent space means an abstract, compressed representation of complex data. We removed this terminology and replaced it with the exact meaning in the work by Geer (2024). They use a three-dimensional vector (latent space) to compress the sea ice emissivity at AMSR2 channels.

Geer, A. J. (2024). Simultaneous inference of sea ice state and surface emissivity model using machine learning and data assimilation. Journal of Advances in Modeling Earth Systems, 16, e2023MS004080. https://doi.org/10.1029/2023MS004080

RC: *4 Figure 6d Simulated TB response: Shown are spectra of undefinded quantities "DeltaTB". Please correct to TB values or define DeltaTB.*

AR: We added a description of $\Delta T_b$ in the figure caption.

RC: *5 Line 418: "Crossing of the warm air intrusion from north to south": Correct to "crossing from north to south of a warm air intrusion" (from south)*

AR: Done.

RC: *6 Line 434: What do you mean with "artificial signals"*

AR: We changed this to "false detections".

RC: *7 Line 439: To avoid misunderstandings, connect tails of linked adjectives in front of a substantive by hyphens, such as "very low wind-slab-correlation lengths". Also elswhere.*

AR: We rephrased the sentence: "The wind slab correlation length is very low within the precipitating system, likely due to the ambiguity with the CLWP signal."

RC: *8 Line 440: Correct or clarify (Fig. C1)*

AR: Changed to "(Appendix C)".

RC: *9 Appendix A: This is very limited information on atmospheric profiles. The limitation to ERA5 data is questionable, here. Warm and humid air inflows into the arctic area can change IWV by an order of magnitude within short time. Therefore, the example with 10% change of a rather dry troposphere is insufficient. Actual water vapour variations could be accounted for, e.g by pattern differences and/or short-time variations at and between 22 and 31 GHz. Also, what I found from ground-based observations is that under advective conditions, the temporal decrease of IWV often corresponds to the precipitation in between.*

AR: We agree that the spatiotemporal variability of IWV is not fully resolved by the 1 h temporal and 31 km spatial resolution of ERA5. Especially, warm air intrusions can exhibit a high variability in both space

and time. However, under most conditions, the spatial gradients and temporal variability are moderate and well-represented by ERA5 as found from the comparison with dropsondes. In principle, the addition of IWV into the state vector would be possible within the optimal estimation framework. Exploiting spatiotemporal patterns might be possible, but there are currently no constraints in space included in the cost function. It should be noted that ideally, more high-frequency channels would be helpful when additionally retrieving IWV, which increases computational cost (Fig. Rev1 and Rev2). In this paper, we decided to focus on the window channels and the relatively stronger CLWP signal, and treat the IWV from ERA5 as true IWV as supported by the dropsonde comparison.

[Figure]

Figure Rev1: Simulated sensitivity of the $T_b$ to changes in integrated water vapor (IWV). In total, 200 a priori profiles are randomly sampled along the *HALO* flight track to capture typical conditions during the campaign.

**RC:** *10 Comment on the specularity parameter s: This parameter is most important at nadir view direction because of the largest difference between lambert scattering and specular reflection at a horizontal surface. On the other hand, there is no polarisation information in this case. Therefore, we need off-nadir observations, too, for real tests of the specularity*

**AR:** Yes, off-nadir observations at horizontal and vertical polarization would be ideal to provide information on specular contributions, i.e., angular and polarization dependence of the emissivity, at each channel. Here, we are limited to the nadir view. These have also been used to assess whether specular or Lambertian reflection is more appropriate under the assumption of smooth emissivity gradients at the wing of absorption lines (e.g., Guedj et al., 2010). Several studies have already shown that Lambertian reflection is more appropriate than specular reflection over snow-covered surfaces (Harlow, 2009; Guedj et al., 2010; Harlow, 2011; Harlow and Essery, 2012; Bormann, 2022). The assumption of pure Lambertian scattering is mainly chosen as the fraction of specular contribution is not known, and the computational cost would double when simulating specular and Lambertian contributions in PAMTRA. Also, the assumption of a fixed fractional specular contribution

[Figure]

Figure Rev2: Same as Fig. Rev1, but expressed as a relative change in IWV of 10 %.

would be a simplification, as it likely depends on snow and ice properties that vary similarly to the emissivity on small spatial scales. This justifies the treatment of the specularity parameter as part of the forward model uncertainty. The mean $T_b$ error due to the error of 0.25 in the specularity parameter depends on the channel as shown in Matzler (2005) and varies from -0.2 K at 22 GHz and -1.7 K at 50 GHz. Thus, potentially larger specular contributions at low frequencies (22 and 31 GHz) would have a minimal impact on the simulation.

Bormann, N.. (2022) Accounting for Lambertian reflection in the assimilation of microwave sounding radiances over snow and sea-ice. Quarterly Journal of the Royal Meteorological Society, 148(747), 2796–2813. Available from: https://doi.org/10.1002/qj.4337

R. C. Harlow, "Millimeter Microwave Emissivities and Effective Temperatures of Snow-Covered Surfaces: Evidence for Lambertian Surface Scattering," in IEEE Transactions on Geoscience and Remote Sensing, vol. 47, no. 7, pp. 1957-1970, July 2009, doi: 10.1109/TGRS.2008.2011893.

R. C. Harlow, "Sea Ice Emissivities and Effective Temperatures at MHS Frequencies: An Analysis of Airborne Microwave Data Measured During Two Arctic Campaigns," in IEEE Transactions on Geoscience and Remote Sensing, vol. 49, no. 4, pp. 1223-1237, April 2011, doi: 10.1109/TGRS.2010.2051555.

Matzler, C. (2005). On the determination of surface emissivity from satellite observations. IEEE Geoscience and remote sensing letters, 2(2), 160-163.